developmental biology, behaviour

digit ratio, 2D : 4D, handgrip strength, women, prenatal testosterone

**Author for correspondence:**
Sonja Windhager
e-mail: sonja.windhager@univie.ac.at

# Handgrip strength and 2D : 4D in women: homogeneous samples challenge the (apparent) gender paradox

Nora Bäck[1], Katrin Schaefer[1,2] and Sonja Windhager[1]

[1]Department of Evolutionary Anthropology, and [2]Human Evolution and Archaeological Sciences (HEAS), University of Vienna, Vienna, Austria

KS, 0000-0002-0173-3092; SW, 0000-0003-1809-8678

The length ratio between the second and the fourth digit (2D : 4D) is a retrospective, non-invasive biomarker for prenatal androgen exposure. It was found to be negatively correlated with handgrip strength (HGS) in men, but the evidence for women is mixed. Such studies in women call for increased detection sensitivity. The present study was designed to reduce potential confounding factors, especially age and ethnicity variation. We measured the digit ratios and HGS of 125 healthy women between 19 and 31 years of age from a remote region in Austria. 2D : 4D of both hands was significantly and negatively correlated with HGS ($n = 125$, right hand: $r = –0.255$, $p = 0.002$, left hand: $r = –0.206$, $p = 0.011$). Size, direction and significance of correlation coefficients remained stable when statistically controlling for age, body weight, body height, body mass index or hours of exercise per week. This yields theory-consistent evidence that HGS and 2D : 4D are clearly associated in women—when sufficiently reducing genetic variation (confounding 2D : 4D), the ontogenetic environment and age ranges (confounding HGS) in the study population. This finding implies similar organizing effects of prenatal androgens as in men, pointing to a more parsimonious developmental mechanism and a new look into its proximate and ultimate causes.

## 1. Introduction

Since pre- and perinatal hormones organize the architecture of the mammalian body and brain [1,2], early exposure to higher levels of testosterone (T) may cause fewer female-like and more male-like characteristics, such as increased muscle mass [3]. The ratio between the second and fourth digits (known as 2D : 4D ratio or digit ratio) of the human hand serves as a marker for sex hormone levels in the fetal environment [4–7] (for a summary of recent criticism see [8]). Also, the difference between the 2D : 4D ratios of the right and left hand (Dr – l) has been suggested as a correlate [9,10], with low 2D : 4D and low Dr – l indicating high prenatal T relative to oestradiol. The 'organizational hypothesis' is supported by many studies that found associations between 2D : 4D ratio and appearance [11–14], behavioural traits [15–18], sexual orientation [19,20] and physical performance [21,22]. For some behaviours, effects of early T exposure may be overruled by activating effects of other hormones within a specific context [23]. Importantly, there is accumulating evidence for 2D : 4D being unrelated to adult T levels [24,25]. However, Crewther & Cook's study in women points towards the possibility of an interaction between 2D : 4D and physical exercise in relation to adult T concentrations, showing that a negative association was present in women with high training hours but not in less physically active women [26]. In this study, we focus on the relationship between 2D : 4D and body strength, specifically handgrip strength (HGS). Multiple studies suggest that 2D : 4D is negatively correlated with HGS in men [27–30] (but see [31] for a null result). This association is consistent over

different ethnic groups [27,32]. In other words, men with a relatively longer ring finger compared to the index finger (lower ratio, higher prenatal T exposure) tend to have higher HGS. The evidence in women is far less conclusive: some suggest that 2D : 4D and HGS are correlated in men, but not in women [28,32,33]. This study investigates this association in female study participants further.

Many studies have focused on men and showed that low digit ratios were associated with better performance in skiing [34], sprinting speed [35], football [36] and male reproductive function [37]. The negative relationship between 2D : 4D and performance in a range of sports, including those that require cardiovascular fitness and strength, was first described by Manning & Taylor [36]. A subsequent consideration of running speeds showed that the effect sizes ranged considerably, with strong associations for middle- and long-distance races (cardiovascular fitness), and weak associations for sprinting (which requires strength) [21,38]. In general, these associations were modified by sex such that correlations were stronger for males than females, while the direction was the same (e.g. [39] for distance running).

Associations between 2D : 4D and HGS were first described by Fink et al. in 2006 for German and Mizos men [27]. These associations were of similar magnitude to that of male sprinting speed. Reports of correlations between 2D : 4D and HGS that included both sexes yielded significant negative associations for men but not for women (for adults [28] and for children [40], but see [41] for a null result). These studies are based on large samples and it is likely that there is in fact a negative relationship between 2D : 4D and HGS in males (both children and adults). However, it may be argued that control for ethnicity is difficult in US samples [28,31]. It is entirely possible that the relationship between 2D : 4D and HGS is indeed negative and real in both men and women, with the former stronger than the latter (as one would expect given that the former experience higher prenatal T than the latter [6]). Nonetheless, if the magnitude of the relationship for men is quite small (e.g. $r$ = about –0.30 in a number of studies [42]), then demonstrating the correlation in women would require that confounding factors such as ethnicity and exercise regime be carefully controlled.

Hone & McCollough [28] and van Anders [33] found no correlation between 2D : 4D and HGS in their European and multi-ethnic female samples. Yet some more recent reports on specific ethnic groups—young Indian women [30], (Han)-Chinese [43], elderly Turkish women [44] and European (presumably Polish) young women [45]—did show a negative relationship of various magnitudes. Moreover, negative correlations have been reported between 2D : 4D ratio and physical fitness/athletic prowess for European female participants [21,22]. This calls for re-examining 2D : 4D and HGS in a more homogeneous European population. This approach would help mitigate potential biases from not controlling for ethnic background [28,33] because digit ratios vary across ethnic groups [27,46–48]. Zhao et al. [32], for example, failed to find a significant correlation in Hani women, albeit probably for another reason: although they included participants from the same village and with the same ethnicity, the age range was broad, suggesting that the lack of correlation was related to documented changes in HGS with age [49–53]. Due to the curvilinear relationship, the statistical analyses (partial correlations) they used might not have efficiently controlled for the effect of age on HGS.

Therefore, we set out to reinvestigate the relationship between 2D : 4D and HGS among women using a design targeted at increasing detection sensitivity by controlling for as many confounding variables as possible. Since female and male foetuses are exposed to the same hormones that merely differ in concentration, we predicted a negative correlation between 2D : 4D ratios and HGS in a controlled group of women.

## 2. Material and methods

### (a) Participants

After excluding 27 participants (for reasons outlined below), the dataset comprised 125 women from the province of Lower Austria with parents of Austrian origin. Most of the women ($n = 121$) resided in the Waldviertel Region, but four women lived in two adjoining regions in Lower Austria. The Waldviertel Region is a flat, structurally weak and sparsely populated upland area in northwestern Lower Austria at the border to the Czech Republic. Until the fall of the Iron Curtain in 1989, the region was relatively secluded, suffering from unemployment and exodus [54]. In spite of the ongoing European integration process, the region has remained somewhat disconnected from the dynamic development elsewhere [55,56].

The parents of 89 participants came exclusively from the Waldviertel. Eight participants had parents from elsewhere in Lower Austria, and another eight had only one parent from Lower Austria. Five participants had either both or one parent from Austrian provinces adjacent to Lower Austria. The remaining 15 participants did not report a specific Austrian region of origin for their parents. Participants were recruited only when between 19 and 31 years of age (mean = 24.2, s.d. = 2.85). Data were collected from July 2019 until October 2019. The described measures were approved by the Ethics Committee of the University of Vienna (reference number 00251). Informed consent was obtained from all participants in the study.

Women with injuries or musculoskeletal disorders were excluded. Participants who practised activities that require high handgrip forces, such as apparatus gymnastics, gymnasium powerlifting, handball and bouldering or climbing, were also not included, as increased HGS was expected [57–59]. Likewise, we excluded women who claimed to take potentially performance-enhancing substances.

### (b) Materials

To measure finger lengths, a mobile A4 scanner (CanoScan LiDE 200) with a manually inserted calibration scale was employed. Scans were made with a resolution of 3306 × 4676 pixel at 400 ppi.

Since HGS is strongly positively correlated with total muscle strength, it can be used as a proxy for overall strength [60]. It can be measured with a handgrip dynamometer, a well-established device suited to examine muscle and health status [61] and to evaluate physical performance of athletes [62]. A mobile hand dynamometer (Jamar Plus +) with an adjustable handle was used to measure HGS in kilogram force. Individual HGS is also influenced by genetic factors [63], diets [64], specific hand exercises [65], age (e.g. [50]), body height, body weight and handedness [49,66].

Questionnaires contained questions about age, body height and body weight of the participant, as these are important determinants of the handgrip evaluation [49]. Participants also answered questions about their origin, occupation, highest completed level of education, parents' highest completed level of education, use of medication, current physical condition, pregnancy status, and the amount and type of sports activities they

**Table 1.** Means, standard deviations, minima, maxima and sample sizes for physical measures of the 125 women from Lower Austria.

|  | mean | s.d. | min | max | *n* |
|---|---|---|---|---|---|
| handgrip strength (kgf) | 33.14 | 5.42 | 16.9 | 44.8 | 125 |
| right 2D : 4D | 0.969 | 0.035 | 0.899 | 1.109 | 125 |
| left 2D : 4D | 0.971 | 0.032 | 0.905 | 1.061 | 125 |
| 2D : 4D right–left difference | −0.002 | 0.023 | −0.060 | 0.066 | 125 |
| age (years) | 24.2 | 2.8 | 19 | 31 | 124 |
| body height (cm) | 166.6 | 5.9 | 152 | 180 | 125 |
| body weight (kg) | 62.5 | 10.9 | 44 | 100 | 124 |
| body mass index (kg m$^{-2}$) | 22.48 | 3.60 | 17.10 | 36.79 | 124 |
| hours of exercise per week | 2.98 | 3.34 | 0.0 | 18.0 | 123 |

engage in. The questionnaire also included a revised version of the Edinburgh Handedness Inventory [67], which was first published by Oldfield [68], to quantify the direction and degree of handedness. More specifically, the eight-item questionnaire (EHI_8)—comprising throwing, writing, toothbrush, scissors, spoon, knife (without fork), striking match, computer mouse—was administered [69].

## (c) Procedure

Women from Waldviertel were recruited via a social media platform (Facebook) and first author's acquaintances (text messages), using a snowball sampling strategy. For the data collection, participants were seated in a quiet, private environment. They were assured that withdrawal was possible at any time and that the collected data were anonymized and used solely for the purpose of this study.

For HGS measurement, body and limb posture were standardized because they are influencing factors for handgrip force [70]. Participants were asked to sit in an upright position, with their measured arm positioned at a 90° angle, their upper arm parallel to the torso and the forearm pointing straight forward. The other arm was supposed to hang down loosely, while the knees were bent and feet placed parallel and flat on the floor. Participants could adjust the handle-width: research shows that the self-selected configuration is optimal for delivering maximum HGS [71]. In a standardized manner, participants were then asked to squeeze as hard as they can. The history of diseases and injuries that could affect strength measurements and finger lengths was recorded by the experimenter. Subsequently, participants were told to answer the questionnaire thoroughly and truthfully.

Then, for the 2D : 4D assessment, the palmar side of both hands was scanned twice with the flatbed scanner. Both hands were positioned on the scanner with all fingers parallel, without hyperextending them or exerting pressure. The tip of the middle finger was aligned with the wrist and elbow. Thereafter, HGS was measured a second time. A debriefing, together with a small chocolate offering, concluded the experiment.

Opposite-gender experimenters enhance physical performance [72,73]. Since the study included only women and was performed by a 24-year-old female experimenter (N.B.), this effect can be ignored.

## (d) Data processing and statistical analysis

In this study, the maximum HGS value (of both hands and measurements) was used for all further analyses. Handedness was operationalized following Veale [74]: Each answer was assigned to a value ('Always right' = 100, 'Usually right' = 50, 'Both equally' = 0, 'Usually left' = −50, 'Always left' = −100).

The mean value of all answers was then calculated to assess handedness (left-handedness: mean < −60, two-handedness: −60 ≤ mean ≤ 60, right-handedness: mean > 60).

Finger length measurements were made from the centre of the most proximal metacarpophalangeal crease to the tip of the digit (fingernails not included). When confronted with difficult shapes or formations of the creases, the instructions of Coates *et al.* [75] were consulted. If measurements differed by more than 0.5 mm, a third measurement was made and the two values closest to each other were chosen for further calculations [76]. Further details and reliability assessments are provided in the electronic supplementary material, S2.

After a Kolmogorov–Smirnov test on normal distribution, the existence of a potential association between HGS and 2D : 4D ratio was analysed by calculating the Pearson correlation coefficient. Additionally, partial correlations were calculated to control for potential effects of age, body height and weight, body mass index and hours of exercise per week. Values of $p < 0.05$ were considered statistically significant. For statistical analysis, the Statistical Package for Social Sciences software (IBM SPSS Statistics v. 26 and 27) was used. Hartigan's dip test statistic for unimodality was calculated using the 'diptest' package [77] in R v. 3.3.2 [78].

## 3. Results

Sample descriptives for the physical measures are given in table 1.

A significant negative correlation between HGS and 2D : 4D was determined in the right ($r = -0.255$, $p = 0.002$, $n = 125$) and the left hand ($r = -0.206$, $p = 0.011$, $n = 125$). Results are visualized in figure 1 and itemized for handedness in table 2.

When splitting the sample by handedness (table 2), right and left 2D : 4D ratios of right-handed participants ($n = 107$) remained significantly and negatively correlated with HGS ($r = -0.259$, $p = 0.004$ and $r = -0.202$, $p = 0.019$, respectively). An even larger negative correlation coefficient (but likely due to the small sample size non-significant) was found in two-handed participants ($n = 13$) for the right and for the left hand ($r = -0.327$, $p = 0.138$ and $r = -0.428$, $p = 0.072$, respectively). The sample size of the five left-handed women was too small to allow interpretation of the statistical values and is reported here for the sake of consistency.

The size, direction and significance of the correlation coefficients remained stable when statistically controlling for age, body weight, body height, body mass index or hours of exercise per week by applying partial correlations

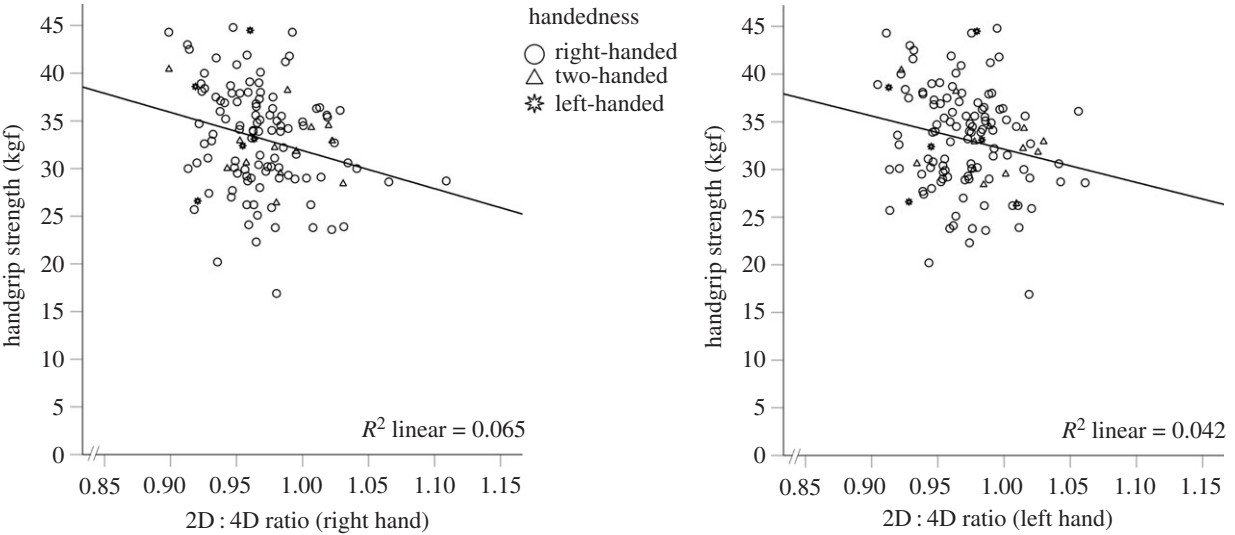

**Figure 1.** Negative relationship between HGS and 2D : 4D. The 2D : 4D ratios of the right and the left hand were significantly ($p < 0.05$) and negatively correlated with HGS ($n = 125$).

**Table 2.** Bivariate correlation between HGS and 2D : 4D ratio split by hand and handedness. All $p$-values are one-tailed and uncorrected.

| (sub-)sample | hand | $n$ | $r$ | $p$ |
|---|---|---|---|---|
| full sample | right | 125 | −0.255 | 0.002 |
| | left | 125 | −0.206 | 0.011 |
| right-handed | right | 107 | −0.259 | 0.004 |
| | left | 107 | −0.202 | 0.019 |
| two-handed | right | 13 | −0.327 | 0.138 |
| | left | 13 | −0.428 | 0.072 |
| left-handed | right | 5 | 0.330 | 0.294 |
| | left | 5 | 0.333 | 0.292 |

(electronic supplementary material, S2), as suggested in previous studies [27,33].

The difference between the 2D : 4D ratios of the right and left hands (Dr − l) generally showed a weaker, negative correlation with HGS, significant only for the right-handed sample (table 3). The direction of association was reversed for the two-handed, but non-significant and limited in interpretation by a sample size of 13. Moreover, for the small samples of two-handed and left-handed participants, the Dr − l values seemed to follow a bimodal distribution based on visual inspection in our sample (electronic supplementary material, S2), which was not confirmed by Hartigan's dip tests (two-handed: $D = 0.093$, $p = 0.399$, $n = 13$; left-handed: $D = 0.161$, $p = 0.204$, $n = 5$), pointing to unimodal distributions.

## 4. Discussion

This study supports the biological hypothesis that higher prenatal T exposure, as approximated by 2D : 4D, and 2D : 4D asymmetry (Dr − l), increases HGS in women. The data analysis showed a significant negative correlation between 2D : 4D and HGS: women with relatively shorter second digits had on average a higher HGS. This association was not confounded by body weight, body height, body mass index, age, handedness or

**Table 3.** Bivariate correlation between HGS and 2D : 4D right–left difference (Dr − l) split by handedness. The correlation is negative between HGS and 2D : 4D for right-handed individuals. Note the low sample sizes and Dr − l distributions for two- and left-handed participants (electronic supplementary material, S2). All $p$-values are one-tailed and uncorrected.

| (sub-)sample | $n$ | $r$ | $p$ | $r_s$ | $p$ |
|---|---|---|---|---|---|
| full sample | 125 | −0.095 | 0.147 | −0.124 | 0.084 |
| right-handed | 107 | −0.111 | 0.128 | −0.167 | 0.042 |
| two-handed | 13 | 0.080 | 0.398 | 0.209 | 0.247 |
| left-handed | 5 | −0.223 | 0.359 | −0.300 | 0.312 |

hours of exercise per week. Furthermore, we found the usual pattern of stronger effect size for right hand 2D : 4D compared to the left [5]. Moreover, our Austrian sample means and standard deviations are in line with other indirectly measured women of European descent (summary table [79]).

Our study confirmed significant negative correlations between 2D : 4D ratio and HGS—when reducing variations in the study population (such as age range, ethnic homogeneity, health status and sporting activity) known to impact 2D : 4D or HGS or both. Our results agree with recent work by Halil *et al.* [44], who reported a comparably moderate correlation coefficient between 2D : 4D ratio and strength in their sample of elderly Turkish women with sarcopenia ($r_s = $ −0.234 and −0.252 for left and right, respectively). Nanda & Samanta [30] found a relationship for the left hands only ($r = $ −0.19) in their sample of young Indian women. Mońka & Pietraszewska [45] reported weaker associations ($r_s$ between −0.10 and −0.15) for their sample of young healthy (presumably Polish) women. Similarly weak negative correlations were reported in young Chinese women of Han ethnicity [43] ($r_s = $ −0.128 and −0.138) as well as of the Ningxia Hui ethnicity [80] ($r_s = $ −0.134 and −0.168). The assumption is that narrowing down the samples to a certain age range and ethnicity helped to increase the signal-to-noise ratio for the association between 2D : 4D and HGS in women.

This assumption is also supported by the null findings of the pioneering studies [28,31–33] with fewer sample

restrictions. Hone & McCullough's (2012) study on psychology students at the University of Miami, USA, showed a trend for a negative, although statistically non-significant, relationship in women ($r = -0.141$, $p = 0.09$) [28]. Since age strongly influences HGS [49,50], and digit ratios vary across ethnic groups [27], controlling for age and ethnic backgrounds seems crucial. Moreover, our results of an even stronger relationship than previously reported (correlation coefficients from $-0.10$ to $-0.14$) support the notion that more such mediating factors are involved, including health status and sporting activity, which might additionally blur the outcome of other studies. Until recently, this was not convincingly accounted for by empirical data. Isen et al. [63] suggested that the lack of a significant association between HGS and 2D : 4D ratio in young women was 'perhaps because individual differences in androgen exposure among typically developing females are too restricted to influence the course of HGS development.' (p. 197).

At first glance, it might be puzzling that male findings seem to be somewhat more robust against sample heterogeneity. Yet, plausible candidates for explanation are (i) the higher prenatal T exposure and HGS in men; (ii) the greater variability in HGS, and maybe also 2D : 4D, in men; and (iii) a steeper regression line in the relationship between 2D : 4D and HGS in men versus women. Males' higher absolute values of T concentrations in utero [6], reflected in a lower 2D : 4D ratio (e.g. [47,81]) and in adult HGS [82,83], might reduce the signal-to-noise ratio in detecting a relationship between 2D : 4D and HGS. Assuming that a regression of HGS upon 2D : 4D yields residuals of the same size in both sexes, then lower variability in women will automatically lead to a lower correlation [84]. Many studies show that men vary more in HGS than women, fitting the picture of greater male variability in many traits [85]. For 2D : 4D the standard deviation is often numerically higher in men than in women, but not significantly so [81]. For a difference in relationship strength (i.e. correlation coefficients), however, the sex difference in HGS variation would be sufficient [84]. A significant interaction term in a regression model in the form of a steeper regression line for men shows that the change of one-unit 2D : 4D will result in a more pronounced change of HGS in men than in women. In other words, early T exposure might have a weaker organizing effect on muscles in women than in men, an effect that has been demonstrated for adult circulating T levels [86]. The article of Hone & McCullough [28] offers all the data to work an example. Men had lower 2D : 4D values than women, indicating higher T exposure. HGS was significantly predicted by sex (men having higher HGS) and the interaction sex * 2D : 4D (steeper regression line in men than women). Their descriptive statistics allowed computing the ratio between male and female variance as well as the corresponding significance test [87]. For HGS, the F statistics equalled 2.16 ($p < 0.001$), indicating that the standard deviation was significantly higher in men than in women (twice as high). For 2D : 4D, the difference between the sexes was non-significant ($F = 0.77$, $p = 0.180$). The results of the worked example are supported by a series of other studies. Greater male HGS as well as HGS variability has also been shown for German samples ([82]; sex difference directly included in the article; variability computed as described above, for the 20–29 year olds and the right hand $F = 2.56$, $p = 0.006$) and Maasai from northern Tanzania [50] (for the age group 20–29, HGS in kgf: male mean = 38.8, s.d. = 8.36, min = 20.0, max = 64.5, $n = 54$; female mean = 26.2,

s.d. = 4.81, min = 17.0, max = 36.5, $n = 41$; $t = 9.23$, $p < 0.001$; $F = 3.02$, $p < 0.001$).

From an evolutionary perspective, physical strength is also sexually dimorphic, with 99.9% of women showing a weaker upper body strength than the average man [88], consistent with reports of higher HGS in men than in women [82,83]. Greater physical strength probably reflects an evolutionary history of male–male competition and physical fighting [89,90]. Also the sexual division of labour in hunter–gatherer societies was named in that context [31]. In a more recent review, Gallup & Fink [91] accumulated evidence for a sex asymmetry in the reported relationships between HGS and measures of social and sexual competition being predominately male-specific. This asymmetry was not found for health status, with HGS being indicative of an individual's health and vitality in both sexes. The suggestion is that sexual selection has led to physiological and cognitive traits that could be advantageous when it comes to intrasexual competition and intersexual choice [92]. Hönekopp et al. [22] argued that sex differences in prenatal T concentrations might be caused by male–male competition. The same holds true for female preferences. Sell, Lukazsweski & Townsley reported that the estimated physical strength determined over 70% of bodily attractiveness in men [93]. However, even if the relationship between 2D : 4D and HGS in men was explained as a legacy of sexual selection, 'it is not immediately apparent why this same proximate mechanism would fail to occur in women', van Anders [33, p. 439] concluded after she was unable to confirm a significant relationship for women, since they are intrauterinely exposed to androgens too, albeit to a lesser extent [6]. This view is supported by findings of negative associations between finger length ratios and women's physical performance, such as sporting success among fencers [94], athletic prowess [21], physical fitness [22], strength performance in athletes [95] and rowing [96].

In a recent meta-analysis, Pasanen et al. [42] found that study-sex-specific correlation coefficients ranged from $-0.62$ to 0.08 (85% negative); overall it was $-0.15$. They concluded that sex did not moderate the relationship between 2D : 4D and HGS. Still, it would be interesting to study men and women under the same rigorous sampling regime to compare male and female effect sizes directly. Such comparisons would also be valuable for understanding the association between 2D : 4D and aerobic exercise (e.g. long-distance running) that generally yields a stronger relationship with 2D : 4D than anaerobic exercise (such as handgrip or sprinting performance) [21,38,42].

Furthermore, on the proximate level, future studies including larger sample sizes altogether, and specifically for both-handed and left-handed individuals, as well as direct measures of body height and body weight could lead to deeper insights into the biological grounds and corroborate the currently available data. Nevertheless, self-reported height and weight are not likely to confound our results to any great extent, as it has been shown that height is on average overreported by a mean of no larger than 2 cm (often less) and weight to be underreported by the same amount in kilograms in many samples of non-obese European women (supplemental tables 1 and 2 of [97]).

The direct link between prenatal testosterone exposure (approximated via 2D : 4D) and adult T levels has been widely dismissed [23,98] (meta-analyses: [24,25]). Crewther & Cook [26] identified 2D : 4D linkages with basal T

and challenge-induced T changes in adult women undertaking regular physical training. They concluded that training hours moderated the 2D:4D link to morning salivary T as well, but with menstrual phase dependency. Further research is needed to determine whether additional physiological layers and environmental relationships may help to understand endocrinological complexities and reconcile inconsistent results.

The 2D:4D link to women's behaviour, and behaviour in general, may be even more plastic and environmentally contingent as compared to physical properties such as muscle strength or facial shape [11–13]. Experimental studies indicate, for example, that the effect of circulating T increase on social behaviour in adult women (e.g. trust, cognitive empathy as tested in economic games) might be moderated by prenatal T [99,100]. Or put differently, the 2D:4D link to behaviour here is sensitive to T activation. In that sense, neurodevelopmental and activational effects interact. Certainly, there might be behaviours for which this is not the case, such as risk taking [23]. Also, many behavioural studies with null effects with regard to 2D:4D and the studied behaviour in the past relied on questionnaires, which may be more susceptible to context and bias as compared to body and face measurements. Accordingly, pre-registered large sample replication studies in all domains of 2D:4D studies would be desirable [101] to separate null relationships from weak relationships not captured by small sample sizes. At the same time, an emphasis on direct behavioural measures, and on interactions with other hormones such as circulating T and cortisol would also be desirable. On a more fundamental note, the empirical evidence for the developmental connection between prenatal T exposure and 2D:4D has, recently, been perceived as insufficient [8]. Indeed, possibilities to study the causal pathways between prenatal T exposure and 2D:4D are constrained for ethical reasons. All this has stirred the scientific debate (for a recent summary of arguments in favour: [20]; opposing views: [8,102]) that

will not be resolved until data availability (including open access to measurements) increases. For a more detailed discussion of the available data sources see electronic supplementary material, S2.

For our part, we showed evidence that when examining homogeneous groups of women, considerable negative correlations between 2D:4D and HGS can be verified, confirming the same pattern in women as in men. This seriously challenges the apparent gender paradox that has long occupied the literature. We hope that we provided the impetus for rethinking the interpretation of the association between 2D:4D ratio and HGS in general. Assuming similar organizing effects of prenatal androgens in both sexes may serve as a more satisfactory and parsimonious developmental explanation. The causes of this variation in prenatal T exposure are a different story, one that is now even more pressing to investigate. Extending the maternal dominance hypothesis [103], intrasexual variation in 2D:4D might reflect a preparation for different life-history strategies relating to the mother's social and environmental circumstances.

Data accessibility. The data are provided in electronic supplementary material [104].

Authors' contributions. N.B.: data curation, formal analysis, investigation, project administration, resources, visualization, writing—original draft, writing—review and editing; K.S.: conceptualization, methodology, project administration, resources, supervision, writing—review and editing; S.W.: conceptualization, data curation, formal analysis, methodology, supervision, writing—review and editing. All authors gave final approval for publication and agreed to be held accountable for the work performed therein.

Competing interests. We declare we have no competing interests.

Funding. We received no funding for this study.

Acknowledgements. We are indebted to our participants for sharing their precious time with us, to Philipp Mitteroecker for thoughts on properties of regressions, to Michael Stachowitsch and to three anonymous reviewers and an associate editor for insightful comments on the text.

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
