## [Peer Review File · Proceedings of the Royal Society B: Biological Sciences]

Review History

RSPB-2021-0307.R0 (Original submission)

Review form: Reviewer 1

Recommendation

Accept with minor revision (please list in comments)

Scientific importance: Is the manuscript an original and important contribution to its field?

Good

General interest: Is the paper of sufficient general interest?

Good

Quality of the paper: Is the overall quality of the paper suitable?

Excellent

Is the length of the paper justified?

Yes

Should the paper be seen by a specialist statistical reviewer?

No

Do you have any concerns about statistical analyses in this paper? If so, please specify them explicitly in your report.

No

It is a condition of publication that authors make their supporting data, code and materials available - either as supplementary material or hosted in an external repository. Please rate, if applicable, the supporting data on the following criteria.

Is it accessible?

N/A

Is it clear?

N/A

Is it adequate?

N/A

Do you have any ethical concerns with this paper?

Yes

Comments to the Author

This is nicely written paper exploring the relationship between 2D:4D digit ratios and hand grip strength (HGS) in females. The authors have designed a study that limits ethnic/genetic, sex, and age confounding factors to test if there is a significant negative correlation between 2D:4D ratio and HGS, as is typically found in males. Previous studies on females have found no correlation or ambiguous results, but have not controlled for all confounding factors, such as age or ethnic background.

This appears to be a robust study with a good sample size (n=125), controlling for parents' origin (rural town in Austria) and age (19-31 yrs). The abstract and introduction are nicely written and informative. The Methods are clearly described. The statistical analyses consider the potential influence of several important parameters (e.g., age, body weight, hours of exercise per week). The results are clearly discussed within the context of previous studies and the evolutionary implications. Overall, it is an impressive piece of research, particularly so as a student Masters project.

Below I outline minor and some more substantial suggestions that I think would improve the quality and clarity of the manuscript. These are listed by line order, rather than by importance.

In the title of manuscript, I think 'solve' is too strong of a word. I don't disagree with arguments the authors lay out in the study or the robusticity of their results, but it is only one study of one population of women in Austria. I would suggest replacing 'solve' with something less declarative.

It might be helpful in the introduction to describe in more layman terms the correlation between the ratio and HGS for novice readers to this topic. e.g., after line 63 "multiple studies suggesting that 2D:4D is negatively correlated with HGS in men", add "men with a relatively longer second digit (higher ratio) tend to have lower HGS". The same might be useful at the start of the Discussion (line 223): e.g. "In other words, women with relatively shorter second digits had on average a higher HGS".

line 80: The following reads a bit awkwardly: "they included participants from the same village and with the same ethnicity but did not narrow down the age range even though HGS changes curvilinearly through lifetime [26, 27, 48-50]." Suggest changing to: "although they included participants from the same village with the same ethnicity, the age range was broad, suggesting

the lack of correlation related to documented changes in HGS with age.”

line 83: In the following sentence, change “Thus, their partial correlations might not have efficiently controlled for the age decline in HGS.” to “Thus, the statistical analyses (partial correlations) they used might not have effectively controlled for the decline in HGS with age”.

line 84: change ‘double-check’ to ‘confirm’ or ‘reinvestigate’

line 87: change to “we predict a negative correlation between 2D:4D ratios...”

line 88: suggest changing to “...in a controlled group of women”.

line 108: change to “Exclusion criteria included women with injuries..”

Remove all italics from Materials and Methods section.

line 119: Were height and weight measured by the first author, or were participants expected to answer this truthfully themselves? If the latter, this may introduce error, particularly for weight, if it is under- or over-estimated by the participant. This could be considered as a limitation of the study at the end of the Discussion section.

Line 124: It would be helpful to provide a bit more detail about the Edinburgh Handedness Inventory for readers not familiar with this questionnaire. How many questions did you use? Did you use the same questionnaire as in Dragovic (2004) that is the “revised version of the Edinburgh Handedness Inventory”?

Lines 128-129: Was recruitment based on an open/public call via Whatsapp/Facebook, or to people already known to the author(s)?

Line 161: Why were the fingerprints blacked out with opaque circles? Was this for anonymity of the data set? Was the fingertip still visible even with contrast enhancement? This part could be more explained and detailed.

Lines 167-168: What are these instructions and how did you deal with the difficulties? Some brief detail would be helpful here.

The subheadings in the Results section are not needed. One or two paragraphs will suffice with reference to the Tables.

line 203: change to “but LIKELY due to small...”

Lines 225-246: This general background is repetitive with the Introduction. I would suggest shifting some of these details to the Introduction and deleting the rest in the Discussion. The results of this study are not discussed until line 247. I would suggest starting with this, and then moving into the broader context of the previous studies.

line 234: change to “required for physical fighting,”

Line 263: “Since age strongly influences HGS [26, 27], and digit ratios vary across ethnic groups [36], controlling for age and ethnic backgrounds seems crucial.” I agree with this statement but it is particularly interesting that these confounding factors are not important for men.

line 261: “Note that the Hone and McCullough article showed a trend for a negative, although statistically nonsignificant, relationship in women” change to “For example, Hone and McCullough’s (DATE) study on XXXX [describe briefly the study or sample] showed a trend for a

negative...”

Line 265: It is unclear what is meant by two magnitude coefficients. Please clarify.

line 268: “Isen and colleagues attempted an explanation of no significant association between handgrip strength and 2D:4D as follows [23]: “Notably, there is no association between 2d:4d and HGS in young women [...],” change to “Isen and colleagues [23] suggested that the lack of significant association between HGS and 2D:4D ratio in young women was “perhaps because...”

Table 2: change “right-handers” to “right-handed” (and the same for ‘two-handers’ and ‘left-handers’)

Review form: Reviewer 2

Recommendation

Major revision is needed (please make suggestions in comments)

Scientific importance: Is the manuscript an original and important contribution to its field?

Good

General interest: Is the paper of sufficient general interest?

Excellent

Quality of the paper: Is the overall quality of the paper suitable?

Good

Is the length of the paper justified?

Yes

Should the paper be seen by a specialist statistical reviewer?

No

Do you have any concerns about statistical analyses in this paper? If so, please specify them explicitly in your report.

No

It is a condition of publication that authors make their supporting data, code and materials available - either as supplementary material or hosted in an external repository. Please rate, if applicable, the supporting data on the following criteria.

Is it accessible?

Yes

Is it clear?

Yes

Is it adequate?

Yes

Do you have any ethical concerns with this paper?

No

Comments to the Author

This report considers the relationship between digit ratio (2D:4D: a putative negative correlate of prenatal testosterone and a positive correlate of prenatal oestrogen) and handgrip strength (HGS) in a sample of Austrian women (n=125). The authors report that 2D:4D of right and left hands was significantly negatively correlated with HGS (right $r = -0.255$, $p = 0.002$, left hand: $r = -0.206$, $p = 0.011$). Significant negative relationships between 2D:4D and HGS were first reported for men. Later, reports included samples of women. In general correlations were stronger for men and weaker (often non-significant) for women. The authors suggest that careful control for ethnicity (most of their sample originated from a rather isolated region of Austria) is necessary to accurately determine the strength of the relationship between 2D:4D and HGS. They conclude, "HGS and 2D:4D are clearly associated in women – when sufficiently reducing genetic variation (confounding 2D:4D), ontogenetic environment, and age ranges (confounding HGS) in the study population."

I think this study does add to our understanding of the relationship between 2D:4D and strength. However, I do have some comments which might help the authors to improve the report:

Introduction

Most of the relevant studies are reported here but the order of report is rather haphazard and, I think, detracts from an understanding of the issues in this literature. I suggest the authors consider the following as a way to organise the Introduction:

The negative relationship between 2D:4D and performance in a range of sports, including those which require cardiovascular fitness and strength was first described by Manning & Taylor (2001). A subsequent consideration of running speeds showed that the effect sizes ranged considerably, with strong associations for middle- and long-distance races (cardiovascular fitness), and weak associations for sprinting (which requires strength) (Manning et al, 2007; Honekopp & Shuster, 2010). In general, these associations were modified by sex such that correlations were stronger for males than females but they were in the same direction (e.g. Longman et al 2012, for distance running). Associations between 2D:4D and HGS were first described by Fink et al in German and Mizos men (2006). They were of similar magnitude to that of male sprinting speed. Reports of correlations between 2D:4D and HGS which included males and females reported significant negative associations for males but not females (for adults Hone & McCullough, 2012 and for children Ranson et al, 2015 but see Georgiev et al, 2017 for a null result). These studies reported large samples and it is likely that there is in fact a negative relationship between 2D:4D and HGS in males (both children and adults). However, it may be argued that control for ethnicity is difficult in US samples (Hone et al). It is entirely possible that the relationship between 2D:4D and HGS is indeed negative and real in both males and females with the former stronger than the latter (as we would expect given the former experiences high prenatal T than the latter). However, if the magnitude of the relationship for males is quite small (e.g. $r = \text{about } -0.30$) then to demonstrate the correlation in females requires that confounding factors such as ethnicity and exercise regime are carefully controlled. That is what this study does.

Methods – digit measurement

The authors have measured digit lengths indirectly from scans. In this regard this Austrian report is of importance. Indirectly measured 2D:4D has been found to be lower than 2D:4D measured directly from the fingers (Manning et al, 2005). This "distortional" effect of scans or photocopies has been found in reports from a number of labs that have considered samples from a number of nation populations. However, Austrian samples considered by Voracek et al have failed to replicate this effect in three reports (see Ribeiro et al, 2016 ; Fink & Manning, 2018 for a consideration of this failure to replicate). I think that the female Austrian means reported here (right .969, left .971) are consistent with typical indirect means from European populations. Note that it is not known whether direct or indirect 2D:4D best reflects the underlying lengths of the phalanges. This needs to be established. Therefore I have no concerns regarding the accuracy of the digit measurements. Rather the present data are pertinent to the possibility that the "Austrian effect" is lab-dependent rather than nation-dependent. In this regard the careful control of ethnicity is also important. It has been known for some time that mean 2D:4D varies according to ethnicity (Manning, 2002) and also by an the direct/indirect measurement effect. The present

study does not mix indirect and direct 2D:4D and the reported means () are consistent with other European indirect means. In addition the 2D:4D SD's of about 0.03 are what one would expect from accurate digit measurement. The authors should comment on all these points, further work is probably needed to clarify this pattern of null replication in the Voracek/ Austrian reports of direct versus indirect 2D:4D.

Results

The reported p values for the correlations in Appendix 1 are one-tailed. Can the authors clarify whether those in Table 2 are one or two-tailed please.

Also of note is that the authors report the usual pattern of stronger effect size for right hand 2D:4D compared to left (Manning et al, 1998). Could the authors also report the relationship between right-left (Dr-l) and HGS. Low Dr-l has been suggested to be a correlate of high prenatal testosterone (Manning et al, 2000; Manning, 2002, p 21-22) and it has been reported to be correlated to a number of traits that may be testosterone dependent and influenced prenatally, such as spermatogenesis and left-handedness (Manning et al, 2000; Manning, 2002, p. 27-30).

In conclusion, this is a study of 2D:4D and HGS in Austrian women that has carefully controlled for confounding factors such as ethnicity, handedness, age and patterns of exercise in the participants. I would like to see some restructuring of the Introduction and further analysis of the data with regard to Dr-l. However, I do think the present report adds valuable data to the 2D:4D and HGS picture.

It also provides much needed data on mean indirect 2D:4D in Austrian women. These latter data clarify further the surprising pattern of null findings reported for direct versus indirect mean Austrian 2D:4D. In this regard further work is necessary to determine the effect of direct/indirect measurement on mean 2D:4D of adult Austrian males (Ribeiro et al, 2016; Fink & Manning, 2018).

References

- Fink, B. and J.T. Manning, Direct versus indirect measurement of digit ratio: New data from Austria and a critical consideration of clarity of report in 2D:4D studies. *Early Hum Dev*, 2018. 127: p. 28-32.
- Fink, B., et al., Digit ratio and hand-grip strength in German and Mizos men: cross-cultural evidence for an organizing effect of prenatal testosterone on strength. *Am J Hum Biol*, 2006. 18(6): p. 776-82.
- Georgiev, A.V., et al., Second-to-fourth digit ratio (2D:4D) is unrelated to measures of somatic reproductive effort among young men from Cebu, the Philippines. *Am J Phys Anthropol*, 2017. 163(3): p. 437-445.
- Hönekopp, J. and M. Schuster, A meta-analysis on 2D:4D and athletic prowess: Substantial relationships but neither hand out-predicts the other. *Personality and Individual Differences*, 2010. 48(1): p. 4-10.
- Hone, L.S. and M.E. McCullough, 2D: 4D ratios predict hand grip strength (but not hand grip endurance) in men (but not in women). *Evolution and Human Behavior*, 2012. 33(6): p. 780-789.
- Longman, D., J.C. Wells, and J.T. Stock, Can persistence hunting signal male quality? A test considering digit ratio in endurance athletes. *PLoS One*, 2015. 10(4): p. e0121560.
- Manning, J.T., *Digit Ratio: a Pointer to Fertility, Behavior and Health*. 2002, New Brunswick New Jersey, London: Rutgers University Press.
- Manning, J.T., L. Morris, and N. Caswell, Endurance running and digit ratio (2D:4D): implications for fetal testosterone effects on running speed and vascular health. *Am J Hum Biol*, 2007. 19(3): p. 416-21.
- Manning, J.T., et al., The ratio of 2nd to 4th digit length: a predictor of sperm numbers and concentrations of testosterone, luteinizing hormone and oestrogen. *Hum Reprod*, 1998. 13(11): p.

3000-4.

Manning, J.T. and R.P. Taylor, Second to fourth digit ratio and male ability in sport: implications for sexual selection in humans. *Evol Hum Behav*, 2001. 22(1): p. 61-69.

Manning, J.T., et al., Photocopies yield lower digit ratios (2D:4D) than direct finger measurements. *Arch Sex Behav*, 2005. 34(3): p. 329-33.

Manning, J.T., et al., The 2nd:4th digit ratio and asymmetry of hand performance in Jamaican children. *Laterality*, 2000. 5(2): p. 121-32.

Ranson, R., G. Stratton, and S.R. Taylor, Digit ratio (2D:4D) and physical fitness (Eurofit test battery) in school children. *Early Hum Dev*, 2015. 91(5): p. 327-31.

Ribeiro, E., et al., Direct Versus Indirect Measurement of Digit Ratio (2D: 4D) A Critical Review of the Literature and

New Data. *Evolutionary Psychology*, 2016. 14(1): p. 1474704916632536.

Review form: Reviewer 3

Recommendation

Reject – article is not of sufficient interest (we will consider a transfer to another journal)

Scientific importance: Is the manuscript an original and important contribution to its field?

Acceptable

General interest: Is the paper of sufficient general interest?

Acceptable

Quality of the paper: Is the overall quality of the paper suitable?

Good

Is the length of the paper justified?

Yes

Should the paper be seen by a specialist statistical reviewer?

No

Do you have any concerns about statistical analyses in this paper? If so, please specify them explicitly in your report.

No

It is a condition of publication that authors make their supporting data, code and materials available - either as supplementary material or hosted in an external repository. Please rate, if applicable, the supporting data on the following criteria.

Is it accessible?

Yes

Is it clear?

Yes

Is it adequate?

Yes

Do you have any ethical concerns with this paper?

No

Comments to the Author

This paper reexamines the link between 2D4D and HGS among women in a homogeneous European sample, attempting to control for a number of variables known to influence or obscure this relationship. The findings support a significant negative correlation between these measures, which is consistent with some past reports on men.

The paper is well-written and the study appears properly executed. My concerns are with (1) the representation of the existing literature on this topic (2D4D and HGS), and (2) the importance of this work to meet the criteria for publication in PRSB.

A growing literature is showing null results when examining behavioral correlates of 2D4D (e.g., Ronay et al., 2018, *Frontiers in Behavioral Neuroscience*; Neyse et al., 2021, *Journal of Economic Behavior and Organization*), and that this measure may not even represent prenatal androgen exposure (Nave et al., 2021 *Hormones and Behavior*) or be used as a marker of circulating testosterone (Kowal et al., 2020 *Scientific Reports*). In addition, the correlation between 2D4D and HGS is mixed even among male samples (Gallup et al., 2007, *Evolution and Human Behavior*).

While the approach by the authors remains interesting, the issues above notwithstanding, I am not sure the current paper/study warrants publication in PRSB. The sample is relatively small, and the contribution to the field is modest, particularly given the recent scrutiny of 2D4D as a biomarker of behavior. I therefore recommend the authors submit this paper, perhaps as a short report, to a more specialized journal following revision to include a more balanced and representative overview of the literature and the implications of this research.

Minor comment: if the authors propose that 2D4D should indeed predict HGS among women consistently when applying similar techniques, they may wish to integrate this perspective with how HGS consistently demonstrates a striking sexual dimorphism in predicting behavior/personality, other aspects of body morphology, and reproductive fitness among men and women (Gallup & Fink, 2020, *Frontiers in Evolutionary Psychology*).

Decision letter (RSPB-2021-0307.R0)

19-Apr-2021

Dear Dr Windhager:

I am writing to inform you that your manuscript RSPB-2021-0307 entitled "Handgrip strength and 2D:4D in women: homogeneous populations solve the (apparent) gender paradox" has, in its current form, been rejected for publication in *Proceedings B*.

This action has been taken on the advice of referees, who have recommended that substantial revisions are necessary. With this in mind we would be happy to consider a resubmission, provided the comments of the referees are fully addressed. However please note that this is not a provisional acceptance.

It was noted in review that the data in the ESM may not be accessible in terms of formatting; if there is a way to amend this by providing more openly (re)usable data please do so on resubmission and in either case address the issue in your resubmission.

Sincerely,

Dr John Hutchinson, Editor

Associate Editor

Board Member: 1

Comments to Author:

Thank you for the opportunity to review this study. Herein the authors examine the correlation between digit length and grip strength in humans, with a particular focus on the interaction of these parameters with gender and ethnicity. As a basic biomechanics scientist working in an ageing research department I encounter rather dry studies of grip strength quite regularly and so I was very interested by the way in which this study examined grip strength in the context of morphology, gender and ethnicity. It provides a more mechanistic understanding of the underlying biology and gives them a platform to discuss a variety of contexts (clinical, ontogeny, physiology, evolutionary) in the discussion. However, while I am positive about this study, both reviewers have highlighted some issues that require attention, particular broadening discussion of the wider literature and a clearer presentation/structure of the MS throughout. Information on ethical approval appears to be absent but must be provided.

Reviewer(s)' Comments to Author:

Referee: 1

Comments to the Author(s)

This is nicely written paper exploring the relationship between 2D:4D digit ratios and hand grip strength (HGS) in females. The authors have designed a study that limits ethnic/genetic, sex, and age confounding factors to test if there is a significant negative correlation between 2D:4D ratio and HGS, as is typically found in males. Previous studies on females have found no correlation or ambiguous results, but have not controlled for all confounding factors, such as age or ethnic background.

This appears to be a robust study with a good sample size (n=125), controlling for parents' origin (rural town in Austria) and age (19-31 yrs). The abstract and introduction are nicely written and informative. The Methods are clearly described. The statistical analyses consider the potential influence of several important parameters (e.g., age, body weight, hours of exercise per week).

The results are clearly discussed within the context of previous studies and the evolutionary implications. Overall, it is an impressive piece of research, particularly so as a student Masters project.

Below I outline minor and some more substantial suggestions that I think would improve the quality and clarity of the manuscript. These are listed by line order, rather than by importance.

In the title of manuscript, I think 'solve' is too strong of a word. I don't disagree with arguments the authors lay out in the study or the robusticity of their results, but it is only one study of one population of women in Austria. I would suggest replacing 'solve' with something less declarative.

It might be helpful in the introduction to describe in more layman terms the correlation between the ratio and HGS for novice readers to this topic. e.g., after line 63 "multiple studies suggesting that 2D:4D is negatively correlated with HGS in men", add "men with a relatively longer second digit (higher ratio) tend to have lower HGS". The same might be useful at the start of the Discussion (line 223): e.g. "In other words, women with relatively shorter second digits had on average a higher HGS".

line 80: The following reads a bit awkwardly: "they included participants from the same village and with the same ethnicity but did not narrow down the age range even though HGS changes curvilinearly through lifetime [26, 27, 48-50]." Suggest changing to: "although they included participants from the same village with the same ethnicity, the age range was broad, suggesting the lack of correlation related to documented changes in HGS with age."

line 83: In the following sentence, change "Thus, their partial correlations might not have efficiently controlled for the age decline in HGS." to "Thus, the statistical analyses (partial correlations) they used might not have effectively controlled for the decline in HGS with age".

line 84: change 'double-check' to 'confirm' or 'reinvestigate'

line 87: change to "we predict a negative correlation between 2D:4D ratios..."

line 88: suggest changing to "...in a controlled group of women".

line 108: change to "Exclusion criteria included women with injuries.."

Remove all italics from Materials and Methods section.

line 119: Were height and weight measured by the first author, or were participants expected to answer this truthfully themselves? If the latter, this may introduce error, particularly for weight, if it is under- or over-estimated by the participant. This could be considered as a limitation of the study at the end of the Discussion section.

Line 124: It would be helpful to provide a bit more detail about the Edinburgh Handedness Inventory for readers not familiar with this questionnaire. How many questions did you use? Did you use the same questionnaire as in Dragovic (2004) that is the "revised version of the Edinburgh Handedness Inventory"?

Lines 128-129: Was recruitment based on an open/public call via Whatsapp/Facebook, or to people already known to the author(s)?

Line 161: Why were the fingerprints blacked out with opaque circles? Was this for anonymity of the data set? Was the fingertip still visible even with contrast enhancement? This part could be more explained and detailed.

Lines 167-168: What are these instructions and how did you deal with the difficulties? Some brief detail would be helpful here.

The subheadings in the Results section are not needed. One or two paragraphs will suffice with reference to the Tables.

line 203: change to “but LIKELY due to small...”

Lines 225-246: This general background is repetitive with the Introduction. I would suggest shifting some of these details to the Introduction and deleting the rest in the Discussion. The results of the this study are not discussed until line 247. I would suggest starting with this, and then moving into the broader context of the previous studies.

line 234: change to “required for physical fighting,”

Line 263: “Since age strongly influences HGS [26, 27], and digit ratios vary across ethnic groups [36], controlling for age and ethnic backgrounds seems crucial.” I agree with this statement but it is particularly interesting that these confounding factors are not important for men.

line 261: “Note that the Hone and McCullough article showed a trend for a negative, although statistically nonsignificant, relationship in women” change to “For example, Hone and McCullough’s (DATE) study on XXXX [describe briefly the study or sample] showed a trend for a negative...”

Line 265: It is unclear what is meant by two magnitude coefficients. Please clarify.

line 268: “Isen and colleagues attempted an explanation of no significant association between handgrip strength and 2D:4D as follows [23]: “Notably, there is no association between 2d:4d and HGS in young women [...],” change to “Isen and colleagues [23] suggested that the lack of significant association between HGS and 2D:4D ratio in young women was “perhaps because...”

Table 2: change “right-handers” to “right-handed” (and the same for ‘two-handers’ and ‘left-handers’)

Referee: 2

Comments to the Author(s)

This report considers the relationship between digit ratio (2D:4D: a putative negative correlate of prenatal testosterone and a positive correlate of prenatal oestrogen) and handgrip strength (HGS) in a sample of Austrian women (n=125). The authors report that 2D:4D of right and left hands was significantly negatively correlated with HGS (right $r = -0.255$, $p = 0.002$, left hand: $r = -0.206$, $p = 0.011$). Significant negative relationships between 2D:4D and HGS were first reported for men. Later, reports included samples of women. In general correlations were stronger for men and weaker (often non-significant) for women. The authors suggest that careful control for ethnicity (most of their sample originated from a rather isolated region of Austria) is necessary to accurately determine the strength of the relationship between 2D:4D and HGS. They conclude, “HGS and 2D:4D are clearly associated in women – when sufficiently reducing genetic variation (confounding 2D:4D), ontogenetic environment, and age ranges (confounding HGS) in the study population.”

I think this study does add to our understanding of the relationship between 2D:4D and strength. However, I do have some comments which might help the authors to improve the report:

Introduction

Most of the relevant studies are reported here but the order of report is rather haphazard and, I think, detracts from an understanding of the issues in this literature. I suggest the authors consider the following as a way to organise the Introduction:

The negative relationship between 2D:4D and performance in a range of sports, including those which require cardiovascular fitness and strength was first described by Manning & Taylor (2001). A subsequent consideration of running speeds showed that the effect sizes ranged considerably, with strong associations for middle- and long-distance races (cardiovascular fitness), and weak associations for sprinting (which requires strength) (Manning et al, 2007;

Honekopp & Shuster, 2010). In general, these associations were modified by sex such that correlations were stronger for males than females but they were in the same direction (e.g. Longman et al 2012, for distance running). Associations between 2D:4D and HGS were first described by Fink et al in German and Mizos men (2006). They were of similar magnitude to that of male sprinting speed. Reports of correlations between 2D:4D and HGS which included males and females reported significant negative associations for males but not females (for adults Hone & McCullough, 2012 and for children Ranson et al, 2015 but see Georgiev et al, 2017 for a null result). These studies reported large samples and it is likely that there is in fact a negative relationship between 2D:4D and HGS in males (both children and adults). However, it may be argued that control for ethnicity is difficult in US samples (Hone et al). It is entirely possible that the relationship between 2D:4D and HGS is indeed negative and real in both males and females with the former stronger than the latter (as we would expect given the former experiences high prenatal T than the latter). However, if the magnitude of the relationship for males is quite small (e.g. $r = \text{about } -.30$) then to demonstrate the correlation in females requires that confounding factors such as ethnicity and exercise regime are carefully controlled. That is what this study does.

Methods – digit measurement

The authors have measured digit lengths indirectly from scans. In this regard this Austrian report is of importance. Indirectly measured 2D:4D has been found to be lower than 2D:4D measured directly from the fingers (Manning et al, 2005). This “distortional” effect of scans or photocopies has been found in reports from a number of labs that have considered samples from a number of nation populations. However, Austrian samples considered by Voracek et al have failed to replicate this effect in three reports (see Ribeiro et al, 2016 ; Fink & Manning, 2018 for a consideration of this failure to replicate). I think that the female Austrian means reported here (right .969, left .971) are consistent with typical indirect means from European populations. Note that it is not known whether direct or indirect 2D:4D best reflects the underlying lengths of the phalanges. This needs to be established. Therefore I have no concerns regarding the accuracy of the digit measurements. Rather the present data are pertinent to the possibility that the “Austrian effect” is lab-dependent rather than nation-dependent. In this regard the careful control of ethnicity is also important. It has been known for some time that mean 2D:4D varies according to ethnicity (Manning, 2002) and also by an the direct/indirect measurement effect. The present study does not mix indirect and direct 2D:4D and the reported means () are consistent with other European indirect means. In addition the 2D:4D SD's of about 0.03 are what one would expect from accurate digit measurement. The authors should comment on all these points, further work is probably needed to clarify this pattern of null replication in the Voracek/ Austrian reports of direct versus indirect 2D:4D.

Results

The reported p values for the correlations in Appendix 1 are one-tailed. Can the authors clarify whether those in Table 2 are one or two-tailed please.

Also of note is that the authors report the usual pattern of stronger effect size for right hand 2D:4D compared to left (Manning et al, 1998). Could the authors also report the relationship between right-left (Dr-l) and HGS. Low Dr-l has been suggested to be a correlate of high prenatal testosterone (Manning et al, 2000; Manning, 2002, p 21-22) and it has been reported to be correlated to a number of traits that may be testosterone dependent and influenced prenatally, such as spermatogenesis and left-handedness (Manning et al, 2000; Manning, 2002, p. 27-30).

In conclusion, this is a study of 2D:4D and HGS in Austrian women that has carefully controlled for confounding factors such as ethnicity, handedness, age and patterns of exercise in the participants. I would like to see some restructuring of the Introduction and further analysis of the data with regard to Dr-l. However, I do think the present report adds valuable data to the 2D:4D and HGS picture.

It also provides much needed data on mean indirect 2D:4D in Austrian women. These latter data clarify further the surprising pattern of null findings reported for direct versus indirect mean Austrian 2D:4D. In this regard further work is necessary to determine the effect of direct/indirect measurement on mean 2D:4D of adult Austrian males (Ribeiro et al, 2016; Fink & Manning, 2018).

References

- Fink, B. and J.T. Manning, Direct versus indirect measurement of digit ratio: New data from Austria and a critical consideration of clarity of report in 2D:4D studies. *Early Hum Dev*, 2018. 127: p. 28-32.
- Fink, B., et al., Digit ratio and hand-grip strength in German and Mizos men: cross-cultural evidence for an organizing effect of prenatal testosterone on strength. *Am J Hum Biol*, 2006. 18(6): p. 776-82.
- Georgiev, A.V., et al., Second-to-fourth digit ratio (2D:4D) is unrelated to measures of somatic reproductive effort among young men from Cebu, the Philippines. *Am J Phys Anthropol*, 2017. 163(3): p. 437-445.
- Hönekopp, J. and M. Schuster, A meta-analysis on 2D:4D and athletic prowess: Substantial relationships but neither hand out-predicts the other. *Personality and Individual Differences*, 2010. 48(1): p. 4-10.
- Hone, L.S. and M.E. McCullough, 2D: 4D ratios predict hand grip strength (but not hand grip endurance) in men (but not in women). *Evolution and Human Behavior*, 2012. 33(6): p. 780-789.
- Longman, D., J.C. Wells, and J.T. Stock, Can persistence hunting signal male quality? A test considering digit ratio in endurance athletes. *PLoS One*, 2015. 10(4): p. e0121560.
- Manning, J.T., *Digit Ratio: a Pointer to Fertility, Behavior and Health*. 2002, New Brunswick New Jersey, London: Rutgers University Press.
- Manning, J.T., L. Morris, and N. Caswell, Endurance running and digit ratio (2D:4D): implications for fetal testosterone effects on running speed and vascular health. *Am J Hum Biol*, 2007. 19(3): p. 416-21.
- Manning, J.T., et al., The ratio of 2nd to 4th digit length: a predictor of sperm numbers and concentrations of testosterone, luteinizing hormone and oestrogen. *Hum Reprod*, 1998. 13(11): p. 3000-4.
- Manning, J.T. and R.P. Taylor, Second to fourth digit ratio and male ability in sport: implications for sexual selection in humans. *Evol Hum Behav*, 2001. 22(1): p. 61-69.
- Manning, J.T., et al., Photocopies yield lower digit ratios (2D:4D) than direct finger measurements. *Arch Sex Behav*, 2005. 34(3): p. 329-33.
- Manning, J.T., et al., The 2nd:4th digit ratio and asymmetry of hand performance in Jamaican children. *Laterality*, 2000. 5(2): p. 121-32.
- Ranson, R., G. Stratton, and S.R. Taylor, Digit ratio (2D:4D) and physical fitness (Eurofit test battery) in school children. *Early Hum Dev*, 2015. 91(5): p. 327-31.
- Ribeiro, E., et al., Direct Versus Indirect Measurement of Digit Ratio (2D: 4D) A Critical Review of the Literature and New Data. *Evolutionary Psychology*, 2016. 14(1): p. 1474704916632536.

Referee: 3

Comments to the Author(s)

This paper reexamines the link between 2D4D and HGS among women in a homogeneous European sample, attempting to control for a number of variables known to influence or obscure this relationship. The findings support a significant negative correlation between these measures, which is consistent with some past reports on men.

The paper is well-written and the study appears properly executed. My concerns are with (1) the representation of the existing literature on this topic (2D4D and HGS), and (2) the importance of this work to meet the criteria for publication in PRSB.

A growing literature is showing null results when examining behavioral correlates of 2D4D (e.g., Ronay et al., 2018, *Frontiers in Behavioral Neuroscience*; Neyse et al., 2021, *Journal of Economic*

Behavior and Organization), and that this measure may not even represent prenatal androgen exposure (Nave et al., 2021 Hormones and Behavior) or be used as a marker of circulating testosterone (Kowal et al., 2020 Scientific Reports). In addition, the correlation between 2D4D and HGS is mixed even among male samples (Gallup et al., 2007, Evolution and Human Behavior).

While the approach by the authors remains interesting, the issues above notwithstanding, I am not sure the current paper/study warrants publication in PRSB. The sample is relatively small, and the contribution to the field is modest, particularly given the recent scrutiny of 2D4D as a biomarker of behavior. I therefore recommend the authors submit this paper, perhaps as a short report, to a more specialized journal following revision to include a more balanced and representative overview of the literature and the implications of this research.

Minor comment: if the authors propose that 2D4D should indeed predict HGS among women consistently when applying similar techniques, they may wish to integrate this perspective with how HGS consistently demonstrates a striking sexual dimorphism in predicting behavior/personality, other aspects of body morphology, and reproductive fitness among men and women (Gallup & Fink, 2020, Frontiers in Evolutionary Psychology).

Author's Response to Decision Letter for (RSPB-2021-0307.R0)

See Appendix A.

RSPB-2021-2328.R0

Review form: Reviewer 2

Recommendation

Accept as is

Scientific importance: Is the manuscript an original and important contribution to its field?

Excellent

General interest: Is the paper of sufficient general interest?

Excellent

Quality of the paper: Is the overall quality of the paper suitable?

Excellent

Is the length of the paper justified?

Yes

Should the paper be seen by a specialist statistical reviewer?

No

Do you have any concerns about statistical analyses in this paper? If so, please specify them explicitly in your report.

No

It is a condition of publication that authors make their supporting data, code and materials available - either as supplementary material or hosted in an external repository. Please rate, if applicable, the supporting data on the following criteria.

Is it accessible?

Yes

Is it clear?

Yes

Is it adequate?

Yes

Do you have any ethical concerns with this paper?

No

Comments to the Author

This revision has considerably improved the original report.

The main thrust of the authors' paper concerns the (apparent) sex difference in the relationship between 2D:4D and hand grip strength (HGS). The literature seems to support a significant negative relationship for men but no relationship for women. The present study finds a significant negative relationship for a sample of Austrian women (n=125). Importantly, the sample is carefully controlled for ethnicity and local population effects. In addition the age range is low and handedness and exercise history are also controlled. Thus, it is argued that given sufficient controls (particularly for ethnicity and other population effects and for age) one should see significant negative relationships between 2D:4D and HGS in both males and females.

The Introduction to the paper now sets out the background more accurately and places the relationship between 2D:4D and HGS within the broader field of 2D:4D and sports. The results are of a pattern which is typical of 2D:4D, i.e. stronger effects for the right hand and an effect for right-left 2D:4D (Dr-l) which is greater than for the left hand 2D:4D. The Discussion now considers the wider field in more detail and there is a reanalysis of other published data. The wider context of the links between 2D:4D and prenatal sex steroids is also addressed.

Importantly, the 2021 meta-analysis of 2D:4D and HGS (Pasanen et al, ref [42]) is considered in the Discussion. Pasanen et al found a mean negative association between 2D:4D and HGS of $r = -.15$ which was not modified by sex. There was considerable heterogeneity in the pattern of effects. I think some at least of this heterogeneity arises from lack of controls for ethnicity and local population effects. Thus Pasanen et al does not render the present report redundant, rather it supports their findings.

Within the context of the relationship between 2D:4D and sports performance the association between 2D:4D and HGS (or muscular fitness after Pasanen et al) shows the lowest effect size. It is similar in magnitude to that of sprinting (as the authors remark in the Introduction). Increases in running distance are associated with marked increases in effect sizes such that for distance running 2D:4D is related to times by approximately $r = .60$. Similar effect sizes are found for sports such as rowing, which also depend on aerobic fitness. Comparisons of male and female effect sizes would be valuable for such sports. Perhaps the authors could make this final point in the Discussion.

Review form: Reviewer 3

Recommendation

Accept with minor revision (please list in comments)

Scientific importance: Is the manuscript an original and important contribution to its field?

Marginal

General interest: Is the paper of sufficient general interest?

Good

Quality of the paper: Is the overall quality of the paper suitable?

Good

Is the length of the paper justified?

Yes

Should the paper be seen by a specialist statistical reviewer?

No

Do you have any concerns about statistical analyses in this paper? If so, please specify them explicitly in your report.

No

It is a condition of publication that authors make their supporting data, code and materials available - either as supplementary material or hosted in an external repository. Please rate, if applicable, the supporting data on the following criteria.

Is it accessible?

Yes

Is it clear?

Yes

Is it adequate?

Yes

Do you have any ethical concerns with this paper?

Yes

Comments to the Author

The authors have sufficient addressed my concerns from initial review, and the paper has improved as a result. However, I am just noticing that there is no mention of informed consent or ethics approval.

Decision letter (RSPB-2021-2328.R0)

01-Nov-2021

Dear Dr Windhager

I am pleased to inform you that your manuscript RSPB-2021-2328 entitled "Handgrip strength and 2D:4D in women: homogeneous samples challenge the (apparent) gender paradox" has been accepted for publication in Proceedings B. Congratulations!!

The referee(s) have recommended publication, but also suggest some minor revisions to your manuscript. Therefore, I invite you to respond to the referee(s)' comments and revise your manuscript. Because the schedule for publication is very tight, it is a condition of publication that you submit the revised version of your manuscript within 7 days. If you do not think you will be able to meet this date please let us know.

Please be sure that ethics/informed consent is fully clear in the MS.

[http://datadryad.org/submit?journalID=RSPB&manu=\(Document not available\)](http://datadryad.org/submit?journalID=RSPB&manu=(Document%20not%20available)) which will take you to your unique entry in the Dryad repository. If you have already submitted your data to dryad you can make any necessary revisions to your dataset by following the above link. Please see <https://royalsociety.org/journals/ethics-policies/data-sharing-mining/> for more details.

Sincerely,

Dr John Hutchinson, Editor

Associate Editor

Board Member

Comments to Author:

Thank you for revising the manuscript according to the suggestions of the reviewers.

Reviewer(s)' Comments to Author:

Referee: 2

Comments to the Author(s).

This revision has considerably improved the original report.

The main thrust of the authors' paper concerns the (apparent) sex difference in the relationship between 2D:4D and hand grip strength (HGS). The literature seems to support a significant negative relationship for men but no relationship for women. The present study finds a significant negative relationship for a sample of Austrian women (n=125). Importantly, the sample is carefully controlled for ethnicity and local population effects. In addition the age range is low and handedness and exercise history are also controlled. Thus, it is argued that given sufficient controls (particularly for ethnicity and other population effects and for age) one should see significant negative relationships between 2D:4D and HGS in both males and females.

The Introduction to the paper now sets out the background more accurately and places the relationship between 2D:4D and HGS within the broader field of 2D:4D and sports. The results are of a pattern which is typical of 2D:4D, i.e. stronger effects for the right hand and an effect for right-left 2D:4D (Dr-I) which is greater than for the left hand 2D:4D. The Discussion now considers the wider field in more detail and there is a reanalysis of other published data. The wider context of the links between 2D:4D and prenatal sex steroids is also addressed.

Importantly, the 2021 meta-analysis of 2D:4D and HGS (Pasanen et al, ref [42]) is considered in the Discussion. Pasanen et al found a mean negative association between 2D:4D and HGS of $r = -.15$ which was not modified by sex. There was considerable heterogeneity in the pattern of effects. I think some at least of this heterogeneity arises from lack of controls for ethnicity and local population effects. Thus Pasanen et al does not render the present report redundant, rather it supports their findings.

Within the context of the relationship between 2D:4D and sports performance the association between 2D:4D and HGS (or muscular fitness after Pasanen et al) shows the lowest effect size. It is similar in magnitude to that of sprinting (as the authors remark in the Introduction). Increases in running distance are associated with marked increases in effect sizes such that for distance

running 2D:4D is related to times by approximately $r = .60$. Similar effect sizes are found for sports such as rowing, which also depend on aerobic fitness. Comparisons of male and female effect sizes would be valuable for such sports. Perhaps the authors could make this final point in the Discussion.

Referee: 3

Comments to the Author(s).

The authors have sufficiently addressed my concerns from initial review, and the paper has improved as a result. However, I am just noticing that there is no mention of informed consent or ethics approval.

Author's Response to Decision Letter for (RSPB-2021-2328.R0)

See Appendix B.

Decision letter (RSPB-2021-2328.R1)

11-Nov-2021

Dear Dr Windhager

I am pleased to inform you that your manuscript entitled "Handgrip strength and 2D:4D in women: homogeneous samples challenge the (apparent) gender paradox" has been accepted for publication in Proceedings B.

Data Accessibility section

Open Access

Paper charges

Sincerely,
Proceedings B
mailto: proceedingsb@royalsociety.org

Appendix A

RESPONSE TO REFEREES

RSPB-2021-0307 Handgrip strength and 2D:4D in women

We greatly appreciate the reviewers' time and attention in reviewing our manuscript. A point-by-point response is provided below (reviewers' comments in italics).

Referee 1

Comments to the Author(s)

This is nicely written paper exploring the relationship between 2D:4D digit ratios and hand grip strength (HGS) in females. The authors have designed a study that limits ethnic/genetic, sex, and age confounding factors to test if there is a significant negative correlation between 2D:4D ratio and HGS, as is typically found in males. Previous studies on females have found no correlation or ambiguous results, but have not controlled for all confounding factors, such as age or ethnic background.

This appears to be a robust study with a good sample size (n=125), controlling for parents' origin (rural town in Austria) and age (19-31 yrs). The abstract and introduction are nicely written and informative. The Methods are clearly described. The statistical analyses consider the potential influence of several important parameters (e.g., age, body weight, hours of exercise per week). The results are clearly discussed within the context of previous studies and the evolutionary implications. Overall, it is an impressive piece of research, particularly so as a student Masters project.

Below I outline minor and some more substantial suggestions that I think would improve the quality and clarity of the manuscript. These are listed by line order, rather than by importance.

In the title of manuscript, I think 'solve' is too strong of a word. I don't disagree with arguments the authors lay out in the study or the robusticity of their results, but it is only one study of one population of women in Austria. I would suggest replacing 'solve' with something less declarative.

We thank the reviewer very much for his*her assessment. The revised title now reads "Handgrip strength and 2D:4D in women: homogeneous samples challenge the (apparent) gender paradox".

It might be helpful in the introduction to describe in more layman terms the correlation between the ratio and HGS for novice readers to this topic. e.g., after line 63 "multiple studies suggesting that 2D:4D is negatively correlated with HGS in men", add "men with a relatively longer second digit (higher ratio) tend to have lower HGS". The same might be useful at the start of the Discussion (line 223): e.g. "In other words, women with relatively shorter second digits had on average a higher HGS".

Done (new lines 53f. and 259f.).

Line 80: The following reads a bit awkwardly: "they included participants from the same village and with the same ethnicity but did not narrow down the age range even though HGS changes curvilinearly through lifetime [26, 27, 48–50]." Suggest changing to: "although they included participants from the same village with the same ethnicity, the age range was broad, suggesting the lack of correlation related to documented changes in HGS with age."

Done (new lines 90f.).

Line 83: In the following sentence, change "Thus, their partial correlations might not have efficiently controlled for the age decline in HGS." to "Thus, the statistical analyses (partial correlations) they used might not have effectively controlled for the decline in HGS with age".

We adopted this suggestion, yet rephrased the sentence to include the curvilinearity of the relationship of age and HGS as a reason for the weak statistical control (new lines 89f.): "Zhao et al. [32], for example, failed to find a significant correlation in Hani women, albeit probably for another reason: although they included participants from the same village and with the same ethnicity, the age range was broad, suggesting that the lack of correlation was related to documented changes in HGS with age [49–53]. Due to the curvilinear relationship, the statistical analyses (partial correlations) they used might not have efficiently controlled for the effect of age on HGS."

Line 84: change 'double-check' to 'confirm' or 'reinvestigate'

Done (new line 95).

Line 87: change to "we predict a negative correlation between 2D:4D ratios..."

Done (new line 97): "Since female and male foetuses are exposed to the same hormones that merely differ in concentration, we predicted a negative correlation between 2D:4D ratios and HGS in a controlled group of women."

Line 88: suggest changing to "...in a controlled group of women".

Done (new line 99).

Line 108: change to "Exclusion criteria included women with injuries..."

We rephrased the sentence so that it now reads: "Women with injuries or musculoskeletal disorders were excluded." (Line 120)

Remove all italics from Materials and Methods section.

All italics are now removed throughout the Materials and Methods section.

Line 119: Were height and weight measured by the first author, or were participants expected to answer this truthfully themselves? If the latter, this may introduce error, particularly for weight, if it is under- or over-estimated by the participant. This could be considered as a limitation of the study at the end of the Discussion section.

Body height and weight were self-reported. We added the following lines to the Discussion (lines 348–355): "Furthermore, on the proximate level, future studies including larger sample sizes altogether, and specifically for both-handed and left-handed individuals, as well as direct measures of body height and body weight could lead to deeper insights into the biological grounds and corroborate the currently available data. Nevertheless, self-reported height and weight are not likely to confound our results to any great extent, as it has been shown that height is on average overreported by a mean of no larger than two centimetres (often less) and weight to be underreported by the same amount in kilograms in many samples of non-obese European women (supplemental tables 1 and 2 of [98])."

Line 124: It would be helpful to provide a bit more detail about the Edinburgh Handedness Inventory for readers not familiar with this questionnaire. How many questions did you use? Did you use the same questionnaire as in Dragovic (2004) that is the "revised version of the Edinburgh Handedness Inventory"?

A more detailed description and an additional reference was inserted in the Material and Methods section (lines 143–145): "More specifically, the 8-item questionnaire (EHI_8) – including throwing, writing, toothbrush, scissors, spoon, knife (without fork), striking match, computer mouse – was administered [69]."

Lines 128–129: Was recruitment based on a open/public call via Whatsapp/Facebook, or to people already known to the author(s)?

Women from Waldviertel were recruited via a social media platform (Facebook) and first author's acquaintances (text messages), using a snowball sampling strategy (lines 148f. in the revised manuscript).

Line 161: Why were the fingerprints blacked out with opaque circles? Was this for anonymity of the data set? Was the fingertip still visible even with contrast enhancement? This part could be more explained and detailed.

Correct. This was done for increased anonymity of the stored data. The fingertip and ridges were still visible. For internal review, please see an example picture below. The corresponding sentence in the Material and Methods section was extended and now reads: "Fingerprints were blacked out with opaque circles to increase anonymity of the stored data. The circles were positioned such that they covered only the most distant phalanx and left the

outline of the finger intact, supporting high data quality for the measurements.” (Lines 181–183)

[Example image of the original dataset for internal review.]

Lines 167–168: What are these instructions and how did you deal with the difficulties? Some brief detail would be helpful here.

The following explanations were added to the manuscript: “Following their approach, we excluded extra creases (not reaching the midline from crease to fingertip, running parallel and distal to the main metacarpophalangeal crease) from our measurements. When encountering horizontal Y-shaped creases, measurements were taken from the center of the Y-pattern. For measurements of difficult crease formations, we applied enhanced image post-processing to further improve sharpness and contrast. The authors also discussed those few ambiguous crease formations to locate optimal measuring points.” (Lines 191–197).

The subheadings in the Results section are not needed. One or two paragraphs will suffice with reference to the Tables.

We removed the subheadings from the revised version of the manuscript.

Line 203: change to “but LIKELY due to small...”

Done (new line 231).

Lines 225–246: This general background is repetitive with the Introduction. I would suggest shifting some of these details to the Introduction and deleting the rest in the Discussion. The results of this study are not discussed until line 247. I would suggest starting with this, and then moving into the broader context of the previous studies.

We consolidated the information in the introduction with some parts of the previous discussion, and moved other parts down the discussion as broader context arising from the specific discussion of our results. The resulting changes to the manuscript might be best seen in the tracked-changes version of the manuscript.

Line 234: change to "required for physical fighting,"

The original sentence was somewhat rephrased in relation to the restructuring of the introduction and discussion. We now use the term physical fighting in line 325: "Greater physical strength probably reflects an evolutionary history of male-male competition and physical fighting [90, 91]"

Line 263: "Since age strongly influences HGS [26, 27], and digit ratios vary across ethnic groups [36], controlling for age and ethnic backgrounds seems crucial." I agree with this statement but it is particularly interesting that these confounding factors are not important for men.

Thank you very much for this remark. We totally agree, although they might be "not *that* important" rather than "*not* important". We thought further along these lines and believe that we found some possible candidates to explain this phenomenon. The following paragraph has been added to the discussion (lines 291–321Miami): "At first glance, it might be puzzling that male findings seem to be somewhat more robust against sample heterogeneity. Yet, plausible candidates for explanation are (a) the higher prenatal testosterone exposure and HGS in men, (b) the greater variability in HGS, and maybe also 2D:4D, in men and (c) a steeper regression line in the relationship between 2D:4D and HGS in men versus women. Males' higher absolute values of testosterone concentrations in utero [6], reflected in a lower 2D:4D ratio (e.g., [47, 82]) and in adult HGS [83, 84], might reduce the signal-to-noise ratio in detecting a relationship between 2D:4D and HGS. Assuming that a regression of HGS upon 2D:4D yields residuals of the same size in both sexes, then lower variability in women will automatically lead to a lower correlation [85]. Many studies show that men vary more in HGS than women, fitting the picture of greater male variability in many traits [86]. For 2D:4D the standard deviation is often numerically higher in men than in women, but not significantly so [82]. For a difference in relationship strength, however, the sex difference in HGS variation would be sufficient [85]. A significant interaction term in a regression model in the form of a steeper regression line for men shows that the change of one-unit 2D:4D will result in a more pronounced change of HGS in men than in women. In other words, early testosterone exposure might have a weaker organizing effect on muscles in women than in men, an effect that has been demonstrated for adult circulating testosterone levels [87]. The article of Hone and colleagues [28] offers all the data to work an example: Men had lower 2D:4D values than women, indicating higher T exposure. HGS was significantly predicted by sex (men having higher HGS) and the interaction sex*2D:4D (steeper regression line in men than women). Their descriptive statistics allowed computing the ratio between male and female variance as well as the corresponding significance test [88]. For HGS, the *F* statistics equalled 2.16 ($p < 0.001$), indicating that the standard deviation was significantly higher in men than in women (twice as high). For 2D:4D, the difference

between the sexes was non-significant ($F= 0.77, p= 0.180$). The results of the worked example are supported by a series of other studies. Greater male HGS as well as HGS variability has also been shown for German samples ([83]; sex difference directly included in the article; variability computed as described above, for the 20–29 year olds and the right hand $F= 2.56, p= 0.006$) and Maasai from Northern Tanzania ([50]; for the age group 20–29, HGS in kgf: male mean= 38.8, $SD= 8.36$, min= 20.0, max= 64.5, $n= 54$; female mean = 26.2, $SD= 4.81$, min= 17.0, max = 36.5, $n= 41$; $t= 9.23, p < 0.001$; $F= 3.02, p < 0.001$)."

Line 261: "Note that the Hone and McCullough article showed a trend for a negative, although statistically nonsignificant, relationship in women" change to "For example, Hone and McCullough's (DATE) study on XXXX [describe briefly the study or sample] showed a trend for a negative..."

Done. The sentence now reads: "Hone and McCullough's (2012) study on psychology students at the University of Miami, USA, showed a trend for a negative, although statistically nonsignificant, relationship in women ($r= -0.141, p= 0.09$) [28]." (Lines 279f.).

Line 265: It is unclear what is meant by two magnitude coefficients. Please clarify.

We rephrased the sentence so that it now reads: "Moreover, our results of an even stronger relationship than previously reported (correlation coefficients from -0.10 to -0.14) support the notion that more such mediating factors are involved, including health status and sporting activity, which might additionally blur the outcome of other studies." (Lines 283–286).

Line 268: "Isen and colleagues attempted an explanation of no significant association between handgrip strength and 2D:4D as follows [23]: "Notably, there is no association between 2d:4d and HGS in young women [...]," change to "Isen and colleagues [23] suggested that the lack of significant association between HGS and 2D:4D ratio in young women was "perhaps because...."

Done (new line 287f.).

Table 2: change "right-handers" to "right-handed" (and the same for 'two-handers' and 'left-handers')

Done.

We thank this reviewer for the thoughtful comments. We hope that we sufficiently addressed all points raised and think this greatly helped to improve the clarity of the manuscript.

Referee 2

Comments to the Author(s)

This report considers the relationship between digit ratio (2D:4D: a putative negative correlate of prenatal testosterone and a positive correlate of prenatal oestrogen) and handgrip strength (HGS) in a sample of Austrian women (n=125). The authors report that 2D:4D of right and left hands was significantly negatively correlated with HGS (right $r = -0.255$, $p = 0.002$, left hand: $r = -0.206$, $p = 0.011$). Significant negative relationships between 2D:4D and HGS were first reported for men. Later, reports included samples of women. In general correlations were stronger for men and weaker (often non-significant) for women. The authors suggest that careful control for ethnicity (most of their sample originated from a rather isolated region of Austria) is necessary to accurately determine the strength of the relationship between 2D:4D and HGS. They conclude, "HGS and 2D:4D are clearly associated in women – when sufficiently reducing genetic variation (confounding 2D:4D), ontogenetic environment, and age ranges (confounding HGS) in the study population."

I think this study does add to our understanding of the relationship between 2D:4D and strength. However, I do have some comments which might help the authors to improve the report:

Introduction

Most of the relevant studies are reported here but the order of report is rather haphazard and, I think, detracts from an understanding of the issues in this literature. I suggest the authors consider the following as a way to organise the Introduction:

The negative relationship between 2D:4D and performance in a range of sports, including those which require cardiovascular fitness and strength was first described by Manning & Taylor (2001). A subsequent consideration of running speeds showed that the effect sizes ranged considerably, with strong associations for middle- and long-distance races (cardiovascular fitness), and weak associations for sprinting (which requires strength) (Manning et al, 2007; Honekopp & Shuster, 2010). In general, these associations were modified by sex such that correlations were stronger for males than females but they were in the same direction (e.g. Longman et al 2012, for distance running). Associations between 2D:4D and HGS were first described by Fink et al in German and Mizos men (2006). They were of similar magnitude to that of male sprinting speed. Reports of correlations between 2D:4D and HGS which included males and females reported significant negative associations for males but not females (for adults Hone & McCullough, 2012 and for children Ranson et al, 2015 but see Georgiev et al, 2017 for a null result). These studies reported large samples and it is likely that there is in fact a negative relationship between 2D:4D and HGS in males (both children and adults). However, it may be argued that control for ethnicity is difficult in US samples (Hone et al). It is entirely possible that the relationship between 2D:4D and HGS is indeed negative and real in both males and females with the former stronger than the latter (as we would expect given the former experiences high prenatal T than the latter). However, if the magnitude of the relationship for males is quite small (e.g. $r =$ about -0.30) then to demonstrate the correlation in females requires that confounding factors such as ethnicity and exercise regime are carefully controlled. That is what this study does.

We thank the reviewer very much for his*her constructive comments. We adopted this suggestion in the introduction of the revised manuscript (second and third paragraph in the

new version). As we stuck very closely to the reviewer's suggestion, the inserts are not repeated in the response letter.

Methods – digit measurement

The authors have measured digit lengths indirectly from scans. In this regard this Austrian report is of importance. Indirectly measured 2D:4D has been found to be lower than 2D:4D measured directly from the fingers (Manning et al, 2005). This "distortional" effect of scans or photocopies has been found in reports from a number of labs that have considered samples from a number of nation populations. However, Austrian samples considered by Voracek et al have failed to replicate this effect in three reports (see Ribeiro et al, 2016 ; Fink & Manning, 2018 for a consideration of this failure to replicate). I think that the female Austrian means reported here (right .969, left .971) are consistent with typical indirect means from European populations.

Thank you very much for pointing this out. We conducted a series of t-tests to compare our means and standard deviations to the results of those studies which included participants of European descent listed in Fink and Manning (2018), one of the most recent review articles. The following sentence has been added to the discussion: "Moreover, our Austrian sample means and standard deviations are in line with other indirectly measured women of European descent (summary table [80])" (new lines 263f.).

Note that it is not known whether direct or indirect 2D:4D best reflects the underlying lengths of the phalanges. This needs to be established. Therefore I have no concerns regarding the accuracy of the digit measurements. Rather the present data are pertinent to the possibility that the "Austrian effect" is lab-dependent rather than nation-dependent. In this regard the careful control of ethnicity is also important. It has been known for some time that mean 2D:4D varies according to ethnicity (Manning, 2002) and also by an the direct/indirect measurement effect. The present study does not mix indirect and direct 2D:4D and the reported means () are consistent with other European indirect means. In addition the 2D:4D SD's of about 0.03 are what one would expect from accurate digit measurement. The authors should comment on all these points, further work is probably needed to clarify this pattern of null replication in the Voracek/Austrian reports of direct versus indirect 2D:4D.

As we only sampled indirect measurements of women, unfortunately, we cannot add much evidence to this interesting discourse about direct vs. indirect 2D:4D male measures in Austria and elsewhere at this point. The only contribution we can think of is a direct comparison between the indirect female Austrian measurements available: Our 2D:4D ratios do not differ significantly from the indirect measurements in Austrian women reported by Voracek & Offenmüller (2007) and Dressler & Voracek (2011), nor do they differ from a sample of Austrian girls and women (Fink & Manning, 2018). A sample of Austrian and British women (Manning et al., 2005) yielded slightly higher values ($p= 0.073$ for the right hand, $p= 0.059$ for the left hand), and Fink et al. (2005) significantly higher values for a sample of undergraduates from the university of Vienna (Austria). Interestingly, the latter two were the only ones using a photocopier, whereas all the others used flatbed scanners. Therefore, a "device effect" cannot be excluded as an explanation for this phenomenon.

Results

The reported p values for the correlations in Appendix 1 are one-tailed. Can the authors clarify whether those in Table 2 are one or two-tailed please.

Thank you very much for pointing to this. The reported *p*-values in Table 2 are one-tailed as well. "All *p*-values are one-tailed and uncorrected." was added to the new Table 2 caption.

Also of note is that the authors report the usual pattern of stronger effect size for right hand 2D:4D compared to left (Manning et al, 1998).

Done. We inserted the following lines: "Furthermore, we found the usual pattern of stronger effect size for right hand 2D:4D compared to the left [79]." (Lines 262f).

Could the authors also report the relationship between right-left (Dr-l) and HGS. Low Dr-l has been suggested to be a correlate of high prenatal testosterone (Manning et al, 2000; Manning, 2002, p 21-22) and it has been reported to be correlated to a number of traits that may be testosterone dependent and influenced prenatally, such as spermatogenesis and left-handedness (Manning et al, 2000; Manning, 2002, p. 27-30).

We inserted a note on this additional proxy for prenatal androgen exposure in the introduction ("Also, the difference between the 2D:4D ratios of the right and left hand (Dr-l) has been suggested as a correlate [9, 10], with low 2D:4D and low Dr-l indicating high prenatal testosterone relative to estradiol.", lines 39-42) and the discussion ("as approximated by 2D:4D, and 2D:4D asymmetry (Dr-l)," line 258). The following paragraph was added to the result section (lines 243-250) together with a new table (Table 3 – listing the correlation coefficients and *p*-values overall and split by handedness) and an electronic supplement (ESM 2): "The difference between the 2D:4D ratios of the right and left hands (Dr-l) generally showed a weaker, negative correlation with HGS, significant only for the right-handed sample (Table 3). The direction of association was reversed for the two-handed, but non-significant and limited in interpretation by a sample size of 13. Moreover, for the small samples of two-handed and left-handed participants, the Dr-l values seemed to follow a bimodal distribution based on visual inspection in our sample (Electronic Supplementary Material 2), which was not confirmed by Hartigan's dip tests (two-handed: $D = 0.093$, $p = 0.399$, $n = 13$; left-handed: $D = 0.161$, $p = 0.204$, $n = 5$) pointing to unimodal distributions.". We also included the descriptive statistics for Dr-l in Table 2.

Table 3. Bivariate correlation between HGS and 2D:4D right-left difference (Dr-l) split by handedness.

A negative correlation between HGS and 2D:4D was found for right-handed individuals. Note the low sample sizes and Dr-l distributions for two- and left-handed participants (Electronic Supplementary Material 2). All *p*-values are one-tailed and uncorrected.

(Sub-)sample	n	r	p	r_s	p
Full sample	125	-0.095	0.147	-0.124	0.084
Right-handed	107	-0.111	0.128	-0.167	0.042
Two-handed	13	0.080	0.398	0.209	0.247
Left-handed	5	-0.223	0.359	-0.300	0.312

In conclusion, this is a study of 2D:4D and HGS in Austrian women that has carefully controlled for confounding factors such as ethnicity, handedness, age and patterns of exercise in the participants. I would like to see some restructuring of the Introduction and further analysis of the data with regard to Dr-l. However, I do think the present report adds valuable data to the 2D:4D and HGS picture. It also provides much needed data on mean indirect 2D:4D in Austrian women. These latter data clarify further the surprising pattern of null findings reported for direct versus indirect mean Austrian 2D:4D. In this regard further work is necessary to determine the effect of direct/indirect measurement on mean 2D:4D of adult Austrian males (Ribeiro et al, 2016; Fink & Manning, 2018).

We thank this reviewer for sharing many details of his*her expertise with us. We hope that we sufficiently addressed all points raised and think this greatly helped to improve the manuscript.

References

- Fink, B. and J.T. Manning, Direct versus indirect measurement of digit ratio: New data from Austria and a critical consideration of clarity of report in 2D:4D studies. *Early Hum Dev*, 2018. 127: p. 28-32.
- Fink, B., et al., Digit ratio and hand-grip strength in German and Mizos men: cross-cultural evidence for an organizing effect of prenatal testosterone on strength. *Am J Hum Biol*, 2006. 18(6): p. 776-82.
- Georgiev, A.V., et al., Second-to-fourth digit ratio (2D:4D) is unrelated to measures of somatic reproductive effort among young men from Cebu, the Philippines. *Am J Phys Anthropol*, 2017. 163(3): p. 437-445.
- Hönekopp, J. and M. Schuster, A meta-analysis on 2D:4D and athletic prowess: Substantial relationships but neither hand out-predicts the other. *Personality and Individual Differences*, 2010. 48(1): p. 4-10.
- Hone, L.S. and M.E. McCullough, 2D: 4D ratios predict hand grip strength (but not hand grip endurance) in men (but not in women). *Evolution and Human Behavior*, 2012. 33(6): p. 780-789.
- Longman, D., J.C. Wells, and J.T. Stock, Can persistence hunting signal male quality? A test considering digit ratio in endurance athletes. *PLoS One*, 2015. 10(4): p. e0121560.
- Manning, J.T., *Digit Ratio: a Pointer to Fertility, Behavior and Health*. 2002, New Brunswick New Jersey, London: Rutgers University Press.
- Manning, J.T., L. Morris, and N. Caswell, Endurance running and digit ratio (2D:4D): implications for fetal testosterone effects on running speed and vascular health. *Am J Hum Biol*, 2007. 19(3): p. 416-21.
- Manning, J.T., et al., The ratio of 2nd to 4th digit length: a predictor of sperm numbers and concentrations of testosterone, luteinizing hormone and oestrogen. *Hum Reprod*, 1998. 13(11): p. 3000-4.

Manning, J.T. and R.P. Taylor, *Second to fourth digit ratio and male ability in sport: implications for sexual selection in humans*. *Evol Hum Behav*, 2001. 22(1): p. 61-69.

Manning, J.T., et al., *Photocopies yield lower digit ratios (2D:4D) than direct finger measurements*. *Arch Sex Behav*, 2005. 34(3): p. 329-33.

Manning, J.T., et al., *The 2nd:4th digit ratio and asymmetry of hand performance in Jamaican children*. *Laterality*, 2000. 5(2): p. 121-32.

Ranson, R., G. Stratton, and S.R. Taylor, *Digit ratio (2D:4D) and physical fitness (Eurofit test battery) in school children*. *Early Hum Dev*, 2015. 91(5): p. 327-31.

Ribeiro, E., et al., *Direct Versus Indirect Measurement of Digit Ratio (2D: 4D) A Critical Review of the Literature and New Data*. *Evolutionary Psychology*, 2016. 14(1): p. 1474704916632536.

Referee 3

Comments to the Author(s)

This paper reexamines the link between 2D:4D and HGS among women in a homogeneous European sample, attempting to control for a number of variables known to influence or obscure this relationship. The findings support a significant negative correlation between these measures, which is consistent with some past reports on men.

The paper is well-written and the study appears properly executed. My concerns are with (1) the representation of the existing literature on this topic (2D:4D and HGS), and (2) the importance of this work to meet the criteria for publication in PRSB.

*A growing literature is showing null results when examining behavioral correlates of 2D:4D (e.g., Ronay et al., 2018, *Frontiers in Behavioral Neuroscience*; Neyse et al., 2021, *Journal of Economic Behavior and Organization*), and that this measure may not even represent prenatal androgen exposure (Nave et al., 2021 *Hormones and Behavior*) or be used as a marker of circulating testosterone (Kowal et al., 2020 *Scientific Reports*).*

We acknowledge the request for giving a broader view of the background. In the original manuscript, we had focused specifically on the relationship between 2D:4D and HGS, leaving aside much of an overall introduction to – and debate on – 2D:4D, as well as its relation to non-physical traits. We now extended the representation of the 2D:4D background, limitations and criticisms in the introduction (lines 39 and 44–50) and in the discussion (lines 356–393): *"The ratio between the second and fourth digit (known as 2D:4D ratio or digit ratio) of the human hand serves as a marker for sex hormone levels in the fetal environment [4–7], for a summary of recent criticism see [8]. [XXX] For some behaviors, effects of early testosterone exposure may be overruled by activating effects of other hormones within a specific context [23]. Importantly, there is accumulating evidence for 2D:4D being unrelated to adult testosterone levels [24, 25]. However, Crewther and Cook's study in women points towards the possibility of an interaction between 2D:4D and physical exercise in relation to adult testosterone concentrations, showing that a negative association was present in women with high training hours but not in less physically active women [26]."* and *"The direct link between prenatal testosterone exposure (approximated via 2D:4D) and adult testosterone levels has been widely dismissed ([99, 100], meta-analyses: [24, 25]). Crewther and Cook [26] identified 2D:4D linkages with basal T and challenge-induced T changes in adult women undertaking regular physical training. They concluded that training hours moderated the 2D:4D link to morning*

salivary testosterone as well, but with menstrual phase dependency. Further research is needed to determine whether adding additional physiological layers and environmental relationships may help to understand endocrinological complexities and reconcile inconsistent results.

The 2D:4D link to women's behaviour, and behaviour in general, may be even more plastic and environmentally contingent as compared to physical properties such as muscle strength or facial shape [11–13]. Experimental studies indicate, for example, that the effect of circulating T increase on social behaviour in adult women (e.g., trust, cognitive empathy as tested in economic games) might be moderated by prenatal T [101, 102]. Or put differently, the 2D:4D link to behaviour here is sensitive to T activation. In that sense, neurodevelopmental and activational effects interact. Certainly, there might be behaviours for which this is not the case, such as risk taking [103]. Also, many behavioural studies with null effects with regard to 2D:4D and the studied behaviour in the past relied on questionnaires, which may be more susceptible to context and bias as compared to body and face measurements. Accordingly, pre-registered large sample replication studies in all domains of 2D:4D studies be desirable [104] to separate null relationships from weak relationships not captured by small sample sizes. At the same time, an emphasis on direct behavioural measures, and on interactions with other hormones such as circulating testosterone and cortisol, would be desirable. On a more fundamental note, the empirical evidence for the developmental connection between prenatal testosterone exposure and 2D:4D has recently been perceived as insufficient [8]. Indeed, possibilities to study the causal pathways between prenatal testosterone exposure and 2D:4D are constrained for ethical reasons. So far, only a limited amount of studies in rodent model organisms are available, two of them confirming a negative association [7, 105], and one with a null result in the front paws corresponding to the human hand [106]. In humans, evidence is limited to amniotic fluid hormonal assessments and postnatal finger measures (pooled sexes [4], significant in girls [107], but see [108]). Studies on genetic disorders, such as congenital adrenal hyperplasia leading to impaired steroidogenesis and increased androgen exposure in female fetuses, are often underpowered, but overall confirm 2D:4D as a retrospective biomarker [109]. It remains unclear whether null findings question the validity of 2D:4D or suffer from sample constraints due to such rare conditions (e.g., Nave et al. [110] present highly unbalanced sample sizes between affected girls/women and controls, large age groups for still growing individuals, mixed ethnicities and only one hand). All this has stirred the scientific debate (for a recent summary of arguments in favour: [20]; opposing views: [8, 111]) that will not be resolved until data availability (including open access to measurements) increases."

However, we only partly agree with this reviewer's interpretation of the literature: In our opinion, the findings on 2D:4D and behavioral traits are not directly generalizable to associations with physical traits (as HGS) because the latter are more plastic and environmentally contingent. Therefore, we feel that we have to refrain from using the correlation between 2D:4D and behavioral traits as an argument against a connection between 2D:4D and HGS.

For the representation of 2D:4D as prenatal testosterone, we have now worked through the current literature and still find more empirical evidence in favour of this concept than against it. More closely examining Nave et al. (2021) for example, we find the sample sizes to be highly unbalanced, the sample composition very special, age ranges large and ethnicities mixed. Thus, we doubt that the source of the "null findings" unequivocally dismisses 2D:4D as a biomarker.

With the final point raised, we completely agree with the reviewer: The direct link between prenatal testosterone exposure (approximated via 2D:4D) and adult testosterone

levels has been widely dismissed. However, this does not preclude organizing effects of prenatal testosterone, nor interaction effects between prenatal and current testosterone (e.g., Crewther and Cook, 2019).

In addition, the correlation between 2D4D and HGS is mixed even among male samples (Gallup et al., 2007, Evolution and Human Behavior).

Yes, we have now extended the review on male samples in the revised manuscript including the reference mentioned above (lines 52f.): “Multiple studies suggest that 2D:4D is negatively correlated with HGS in men [27–30], but see [31] for a null result.” The argument is also picked up in lines 70–75 of the revised manuscript: “Reports of correlations between 2D:4D and HGS which included both sexes reported significant negative associations for men but not for women (for adults [28] and for children [40], but see [41] for a null result). These studies are based on large samples and it is likely that there is in fact a negative relationship between 2D:4D and HGS in males (both children and adults). However, it may be argued that control for ethnicity is difficult in US samples [28, 31].”

While the approach by the authors remains interesting, the issues above notwithstanding, I am not sure the current paper/study warrants publication in PRSB. The sample is relatively small, and the contribution to the field is modest, particularly given the recent scrutiny of 2D4D as a biomarker of behavior. I therefore recommend the authors submit this paper, perhaps as a short report, to a more specialized journal following revision to include a more balanced and representative overview of the literature and the implications of this research.

We thank this reviewer for the significant input to discuss the topic in a broader context. We hope that we were able to resolve most of this reviewer’s concerns.

Minor comment: if the authors propose that 2D4D should indeed predict HGS among women consistently when applying similar techniques, they may wish to integrate this perspective with how HGS consistently demonstrates a striking sexual dimorphism in predicting behavior/personality, other aspects of body morphology, and reproductive fitness among men and women (Gallup & Fink, 2020, Frontiers in Evolutionary Psychology).

Even after an extended literature search (search machines, journal homepage, author homepage), we could not identify any such article from 2020. We therefore assume that this comment refers to the 2018 review of these two authors. The sex asymmetry for HGS correlating with many measures of social and sexual competition is now highlighted in the discussion (lines 327–330): “In a more recent review, Gallup and Fink [92] accumulated evidence for a sex asymmetry in the reported relationships between HGS and measures of social and sexual competition being predominately male-specific. This asymmetry was not found for health status, with HGS being indicative of an individual’s health and vitality in both sexes.”

We thank the reviewer very much for pointing this out.

Appendix B

RESPONSE TO REFEREES

RSPB-2021-2328 Handgrip strength and 2D:4D in women

We greatly appreciate the reviewers' time and attention in reviewing our manuscript. A point-by-point response is provided below (reviewers' comments in italics).

Referee 2

Comments to the Author(s)

This revision has considerably improved the original report.

The main thrust of the authors' paper concerns the (apparent) sex difference in the relationship between 2D:4D and hand grip strength (HGS). The literature seems to support a significant negative relationship for men but no relationship for women. The present study finds a significant negative relationship for a sample of Austrian women (n=125). Importantly, the sample is carefully controlled for ethnicity and local population effects. In addition the age range is low and handedness and exercise history are also controlled. Thus, it is argued that given sufficient controls (particularly for ethnicity and other population effects and for age) one should see significant negative relationships between 2D:4D and HGS in both males and females.

The Introduction to the paper now sets out the background more accurately and places the relationship between 2D:4D and HGS within the broader field of 2D:4D and sports. The results are of a pattern which is typical of 2D:4D, i.e. stronger effects for the right hand and an effect for right-left 2D:4D (Dr-L) which is greater than for the left hand 2D:4D. The Discussion now considers the wider field in more detail and there is a reanalysis of other published data. The wider context of the links between 2D:4D and prenatal sex steroids is also addressed. Importantly, the 2021 meta-analysis of 2D:4D and HGS (Pasanen et al, ref [42]) is considered in the Discussion. Pasanen et al found a mean negative association between 2D:4D and HGS of $r = -.15$ which was not modified by sex. There was considerable heterogeneity in the pattern of effects. I think some at least of this heterogeneity arises from lack of controls for ethnicity and local population effects. Thus Pasanen et al does not render the present report redundant, rather it supports their findings.

Within the context of the relationship between 2D:4D and sports performance the association between 2D:4D and HGS (or muscular fitness after Pasanen et al) shows the lowest effect size. It is similar in magnitude to that of sprinting (as the authors remark in the Introduction). Increases in running distance are associated with marked increases in effect sizes such that for distance running 2D:4D is related to times by approximately $r = .60$. Similar effect sizes are found for sports such as rowing, which also depend on aerobic fitness. Comparisons of male and female effect sizes would be valuable for such sports. Perhaps the authors could make this final point in the Discussion.

We adopted the suggestion, so that the corresponding part in the discussion section now reads (lines 349ff.): “Still, it would be interesting to study men and women under the same rigorous sampling regime to compare male and female effect sizes directly. Such comparisons would also be valuable for understanding the association between 2D:4D and aerobic exercise (e.g., long-distance running) that generally yields a stronger relationship with 2D:4D than anaerobic exercise (such as handgrip or sprinting performance) [21, 38, 42].”

Referee 3

Comments to the Author(s)

The authors have sufficient addressed my concerns from initial review, and the paper has improved as a result. However, I am just noticing that there is no mention of informed consent or ethics approval.

A statement on ethics approval was already part of the manuscript. We now extended this part to state that informed consent was obtained from each participant explicitly (lines 118–120): “The described measures were approved by the Ethics Committee of the University of Vienna (reference number 00251). Informed consent was obtained from all participants in the study.”